# Orphan receptor GPR153 facilitates vascular damage responses by modulating cAMP levels, YAP/TAZ signaling, and NF-κB activation

Jingchen Shao [1], Jeonghyeon Kwon [1], Tianpeng Wang [1], Stefan Günther [2,3,4], Lukas S. Tombor [3,4,5], Timothy Warwick [3,4,6], Zaib Shaheryar [7], Ralf P. Brandes [3,4,6], Stefanie Dimmeler [3,4,5], Jan Wenzel [7], Stefan Offermanns [1,3,4,8], Markus Schwaninger [7] & Nina Wettschureck [1,3,4,8] ✉

Vascular cells express various G-protein-coupled receptors (GPCRs) with yet unknown function, among them orphan receptor GPR153. GPR153 was upregulated in smooth muscle cells (SMCs) in response to injury, and knockdown of GPR153 resulted in reduced proliferation and mildly altered differentiation in human SMCs. Mice with tamoxifen-inducible, SMC-specific GPR153 deficiency were partially protected against ligation-induced neointima formation, and their SMCs were characterized by reduced proliferation and dedifferentiation. Mechanistically, we show that GPR153 negatively regulates cellular cAMP levels, and thus the absence of GPR153 leads to an increase in CREB phosphorylation, reduced YAP/TAZ levels, and diminished NF-κB activation. Interestingly, a similar role of GPR153 was observed in endothelial cells (ECs), where loss of GPR153 resulted in reduced inflammatory gene expression and protected mice with EC-specific GPR153 deficiency in models of neuroinflammation and stroke. Taken together, orphan receptor GPR153 facilitates pro-inflammatory and pro-proliferative gene expression in ECs and SMCs by controlling cellular cAMP levels, thereby contributing to inflammation and vascular remodeling.

Vascular cells play an important role in the local response to infectious, metabolic, mechanic or hypoxic damage. One of the earliest reactions is the inflammatory activation of endothelial cells (ECs), for example triggered by damage- or pathogen-associated molecular patterns, cytokines, oxidized lipoproteins, or oscillatory shear stress[1]. Activation of endothelial nuclear factor-κB (NF-κB) signaling results in increased expression of adhesion molecules, cytokines and chemokines, thereby facilitating leukocyte recruitment from blood and supporting the

[1]Department of Pharmacology, Max Planck Institute for Heart and Lung Research, Bad Nauheim, Germany. [2]Deep sequencing Platform, Max Planck Institute for Heart and Lung Research, Bad Nauheim, Germany. [3]German Center for Cardiovascular Research (DZHK), Partner Site Rhine-Main, Frankfurt am Main, Germany. [4]Cardiopulmonary Institute, Goethe University, Frankfurt am Main, Germany. [5]Institute of Cardiovascular Regeneration, Center of Molecular Medicine, Goethe University, Frankfurt am Main, Germany. [6]Institute for Cardiovascular Physiology, Goethe University, Frankfurt am Main, Germany. [7]Department of Pharmacology, University of Lübeck, Lübeck, Germany. [8]Centre for Molecular Medicine, Medical Faculty, Goethe University, Frankfurt am Main, Germany. ✉e-mail: Nina.Wettschureck@mpi-bn.mpg.de

inflammatory response[2,3]. Furthermore, local growth factors and cytokines may induce activation of smooth muscle cells (SMCs) in the vascular media, causing a switch from the contractile phenotype to a synthetic phenotype characterized by increased proliferation, migration, and production of extracellular matrix[4,5]. Together with local activation of platelets and fibroblasts, these responses facilitate pathogen clearance and tissue repair, but may also contribute to chronic inflammation and pathological remodeling.

G-protein-coupled receptors (GPCRs) play important regulatory roles in the activation of all these cell types. They are the largest family of transmembrane receptors in eukaryotes and transmit signals of numerous neurotransmitters, hormones, local mediators, or metabolic cues[6]. GPCRs mediate their intracellular effects through activation of heterotrimeric G-proteins of the families $G_s$, $G_{i/o}$, $G_{q/11}$, and $G_{12/13}$[7], and dysregulation of GPCR signaling has been implicated in the pathogenesis of many diseases[8]. In ECs, GPCRs are central regulators of barrier stability, angiogenic behavior, and inflammatory activation. The $G_{q/11}$-coupled receptors for pro-inflammatory factors thrombin and bradykinin, for example, induce actomyosin contraction and VE-cadherin relocalization, resulting in opening of the endothelial barrier for fluids[3,9]. Furthermore, activation of these receptors induces inflammatory gene expression via NF-κB-dependent pathways[3,9]. In contrast, $G_s$-coupled receptors such as the adrenomedullin receptor stabilize the endothelial barrier though Epac1-dependent RAP1 activation and antagonize NF-κB activation[3,9].

In SMCs, GPCRs are crucial regulators of contractility. Vasoconstrictors such as norepinephrine, thromboxane A2, or angiotensin II act through their respective $G_{q/11}$- and $G_{12/13}$- coupled receptors to facilitate actomyosin contractility, whereas $G_s$-coupled GPCRs such as the prostacyclin receptor IP induce relaxant effects[10–12]. In addition, GPCRs play a central role in the regulation of SMC differentiation[11]. In response to different types of injury, mature SMCs can dedifferentiate to the synthetic phenotype, a process that is crucial for wound healing, but also contributes to adverse remodeling in the context of atherosclerosis or neointima formation after angioplasty[4,5]. Dedifferentiating SMCs can adopt distinct phenotypes depending on the pathophysiological context, for example fibroblast-like, macrophage-like, or osteochondrogenic phenotypes[5,13]. The mechanisms regulating SMC (de-) differentiation are complex. A key role plays transcription factor serum response factor (SRF), which interacts with co-activators of the myocardin family to support contractile gene expression[14,15]. Binding of SRF to co-activators of the ternary complex factor family, however, has opposite effects and facilitates dedifferentiation[14,15]. These two types of SRF coactivators are in turn regulated by numerous signaling cascades: Mitogen-activated protein kinases (MAPK), phosphatidylinositol 3-kinase (PI3K)/Akt, Yes-associated protein (YAP), transcriptional co-activator with PDZ-binding motif (TAZ), or NF-κB rather support dedifferentiation[16–19], whereas the differentiated state is facilitated by cAMP/CREB-, RhoA-, and transforming growth factor TGFβ-dependent signaling[15,19,20]. Many of these signaling cascades are modulated by GPCRs. Activation of $G_{q/11}$-coupled receptors promotes proliferation and dedifferentiation through MAPK-dependent phosphorylation of ternary complex factors, while activation of the $G_{12/13}$-coupled receptors promotes expression of contractile proteins through RhoA-dependent nuclear translocation of myocardin-related transcription factors (MRTFs)[11,14,15]. Furthermore, $G_{i/o}$-coupled receptors such as the sphingosine-1-phosphate receptor $S1P_1$ promote SMC proliferation and migration through inhibition of the cAMP/PKA/CREB pathway, whereas $G_s$-coupled receptors like the prostanoid receptors IP, EP2 and EP4 mediate anti-migratory and anti-proliferative effects by enhancing cAMP signaling[11].

In addition to the well-studied GPCRs mentioned above, SMCs and ECs express numerous GPCRs with yet unknown function and ligand, so-called orphan GPCRs[21,22]. We previously investigated changes in GPCR expression in individual ECs and SMCs from healthy mice and mice with acute or chronic inflammation using single-cell expression analyses[23,24]. These analyses showed that numerous orphan GPCRs are upregulated both in ECs and SMCs in response to damage, but in most cases the functional consequences are unknown. One of these receptors is GPR153, a class A orphan receptor that is highly expressed in brain regions such as thalamus, cerebellum and arcuate nucleus[25]. Intracerebroventricular application of antisense nucleotides against GPR153 resulted in a mild reduction in food intake and modest behavioral abnormalities in rats[25], but there are no mechanistic data or knockout models available for GPR153. Outside the CNS, GPR153 is broadly expressed in mesenchymal cells such as SMCs and fibroblasts, but also here its function is unknown.

In this study, we show that GPR153 facilitates NF-κB and YAP/TAZ-dependent pro-inflammatory and pro-proliferative gene expression by controlling cellular cAMP levels both in SMCs and ECs, thereby contributing to neuroinflammation and vascular remodeling.

## Results

### Damage-induced upregulation of orphan GPCRs in murine and human SMCs

To characterize the expression of orphan GPCRs in SMCs under physiological and pathological conditions, we re-analyzed previously published murine single-cell expression data obtained in models of cardiovascular damage. Carotid SMCs from mice subjected to wire injury-induced neointima formation showed elevated expression frequencies for a number of orphan GPCRs (Fig. 1A, reanalyzed from[26]), and the same was true for aortic SMCs from atherosclerotic mice (Fig. 1B, reanalyzed from[23]) or coronary SMCs after induction of myocardial infarction (Fig. 1C, reanalyzed from[27]). In these murine models, GPR153 showed the most prominent and most consistent upregulation, and also in human SMCs harvested from atherosclerotic arteries (Fig. 1D, reanalyzed from[28]) or infarcted human hearts (Fig. 1E, reanalyzed from[29]), GPR153 expression was increased.

### Function of GPR153 in cultured human SMCs

Since damage-induced upregulation of GPR153 might point to a regulatory role in SMC dedifferentiation, we studied the function of this receptor in human aortic SMCs (hAoSMCs). Knockdown of GPR153 using endonuclease-prepared siRNA (esiRNA) was efficient, but did not affect expression of contractile markers such as α smooth muscle actin (αSMA, encoded by *ACTA2*), SM22α (encoded by *TAGLN*), or myosin heavy chain 11 (*MYH11*) on RNA or protein level (Fig. 2A–C). Also expression of inflammatory markers such as intercellular adhesion molecule 1 (*ICAM1*), vascular adhesion molecule 1 (*VCAM1*), or *IL6* was not significantly altered (Fig. 2A–C). Furthermore, we did not find changes in markers for macrophage-like (*CD68*, integrin α-M/*ITGAM*), fibroblast-like (lumican/*LUM*, biglycan/*BGN*) or osteochondrogenic (osteopontin/*SPP1*, alkaline phosphatasep/*ALPL*) phenotypes (Fig. 2D)[5]. However, proliferation markers such as proliferating cell nuclear antigen (*PCNA*) and marker of proliferation Ki-67 (*MKI67*) were reduced in GPR153 knockdown hAoSMCs (Fig. 2E), and cell number expansion was delayed both under basal culture conditions and after additional growth stimulation with platelet-derived growth factor (PDGF) BB (Fig. 2F). Furthermore, incorporation of 5-ethynyl-2′-deoxyuridine (EdU) was reduced in GPR153 knockdown hAoSMCs (Fig. 2G), and the percentage of annexin V-positive apoptotic cells was increased (Fig. 2H). The overall viability of GPR153-deficient hAoSMCs, however, was not decreased (Fig. 2I). Also in GPR153-deficient human coronary artery SMCs (hCASMCs), proliferation markers were reduced, in this case accompanied by reduced *ACTA2* expression (Fig. 2J). An independent siRNA confirmed reduced proliferation in GPR153-deficient hCASMCs (Fig. 2K, L).

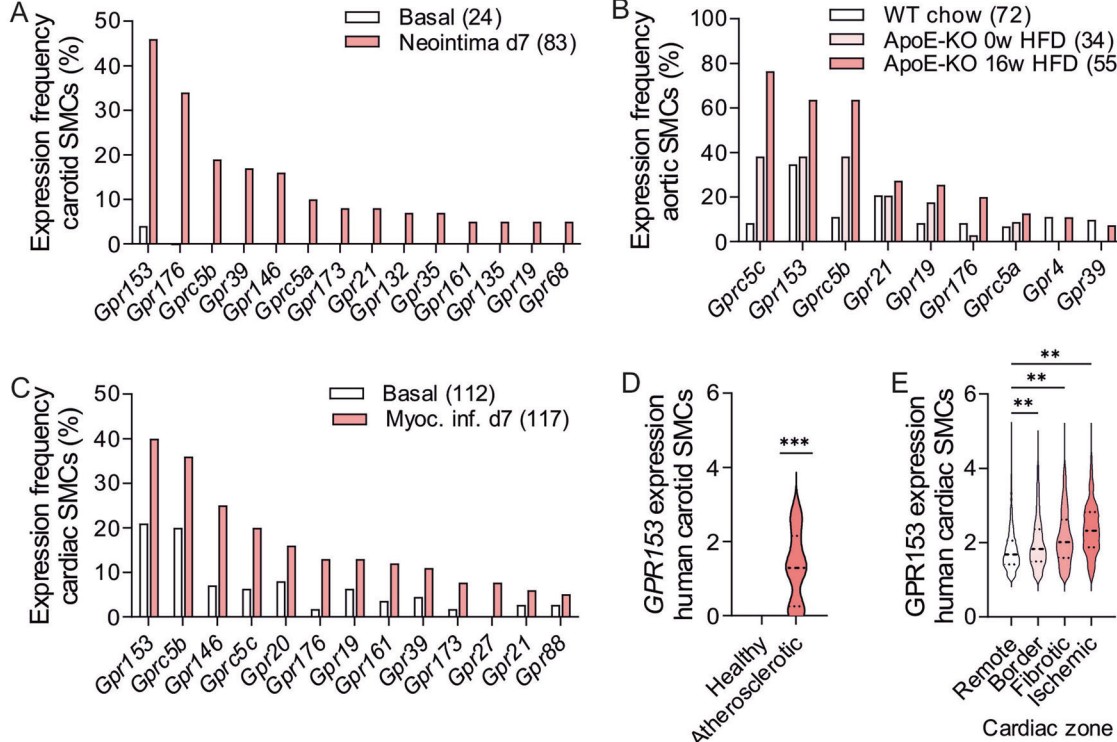

**Fig. 1 | Damage-induced upregulation of orphan GPCRs in murine and human SMCs. A–C** Percentage of murine SMCs expressing specific orphan GPCRs under basal conditions and after induction of neointima formation (**A**, from ref. 26), atherosclerosis (**B**, from ref. 23), or myocardial infarction (**C**, from ref. 27). Adhesion and frizzled GPCRs were omitted; total numbers of cells per group are given in brackets. **D, E** Strength of GPR153 expression in SMCs from human carotid arteries harvested from healthy or atherosclerotic donors (**D**, from ref. 28) or in different areas of infarcted human hearts (**E**, from ref. 29). Only GPR153-positive cells are shown, hatched lines indicate 25th percentile, median, and 75th percentile; differences between groups in (**D**, **E**) were analyzed by two-sided one-sample *t* test (**D**) or one-way ANOVA and Dunnett's post hoc test (**E**). d days after intervention; **P < 0.01; ***P < 0.001.

## Generation and characterization of tamoxifen-inducible, SMC-specific GPR153 knockout mice

To study the role of GPR153 in SMCs in vivo, we generated *Gpr153*-flox mice using CRISPR/Cas9 genome editing (Fig. 3A) and bred them to *Myh11*-CreERT2 mice[30], which allow tamoxifen-inducible expression of recombinase Cre specifically in SMCs. Quantitative reverse transcription polymerase chain reaction (qRT-PCR) showed a clear reduction of *Gpr153* transcripts in the aortic media of tamoxifen-treated *Myh11*-CreERT2; *Gpr153*^fl/fl mice (iSM-G153-KO) (Fig. 3B), and in line with our findings in human SMCs, EdU incorporation in cultured aortic SMCs was reduced (Fig. 3C, D). Though *Gpr153* was poorly detected by single-cell RNA sequencing in carotid arteries of healthy mice (Fig. 1A), qRT-PCR readily detected *Gpr153* mRNA in the cleaned media of control mice and confirmed efficient knockout in iSM-G153-KOs (Fig. 3E). Also in these vessels, expression of proliferation markers *Mki67* and *Pcna* was reduced (Fig. 3F). Interestingly, we also detected significantly reduced expression of contractile markers such as *Myh11* and *Acta2* on the RNA level (Fig. 3F), but on the protein level αSMA or SM22α were only in trend reduced (Fig. 3G, H). To test whether these changes affected vascular tone in iSM-G153-KO arteries, we studied vascular contractility in wire myography. We did not find differences in the responsiveness to contractile stimuli such as the α1 adrenergic agonist phenylephrine or the thromboxane A₂ analog U46619 (Fig. 3I, J). Also responses to the nitric oxide donor sodium nitroprusside, a direct SMC relaxant, or endothelium-dependent relaxation induced by acetylcholine were not significantly changed in knockout vessels, though there was a trend to enhanced responsiveness to cicaprost, an agonist at the G$_s$-coupled prostacyclin receptor (Supplementary Fig. 1A–C). Telemetric analyses of basal blood pressure did not reveal differences between the genotypes, neither in the basal state nor after induction of hypertension by implantation of angiotensin II-releasing miniosmotic

pumps (Supplementary Fig. 1D). Taken together, inactivation of GPR153 in SMCs does not substantially affect basic vascular function, but results in significantly reduced proliferation and mildly altered expression of contractile markers both in human and murine SMCs.

## GPR153 regulates SMC proliferation and dedifferentiation in response to vascular damage

Because reduced SMC proliferation and altered expression of contractile markers might become relevant upon injury-induced SMC dedifferentiation, we studied neointima formation in iSM-G153-KOs in the carotid artery ligation model. 28 days after ligation, iSM-G153-KOs showed reduced neointima formation (Fig. 4A–C), whereas media thickness did not differ (Fig. 4D). QRT-PCR analysis in the media of carotid arteries harvested on day 7 after ligation showed that proliferation marker *Mki67* was upregulated in control samples, but not in KO samples (Fig. 4E). In line with this, EdU incorporation was reduced in the media of ligated carotid arteries harvested from iSM-G153-KOs on d14 after ligation (Fig. 4F, G). Furthermore, contractile markers *Acta2* and *Myh11* were downregulated in control samples, but this effect was reduced in KOs (Fig. 4H, I). Inflammation markers such as *Vcam1* and *Il6* were induced in carotid arteries of control mice, but less so in KOs (Fig. 4J, K). Taken together, reduced neointima formation in iSM-G153-KOs is accompanied by diminished induction of markers for proliferation and inflammation, and by reduced downregulation of contractile genes.

## GPR153 regulates CREB, YAP/TAZ, and NF-κB signaling in cultured SMCs

To understand how GPR153 regulates SMC function, we studied signaling cascades related to SMC proliferation and differentiation in human SMC. Because siRNA-mediated knockdown in hCASMC

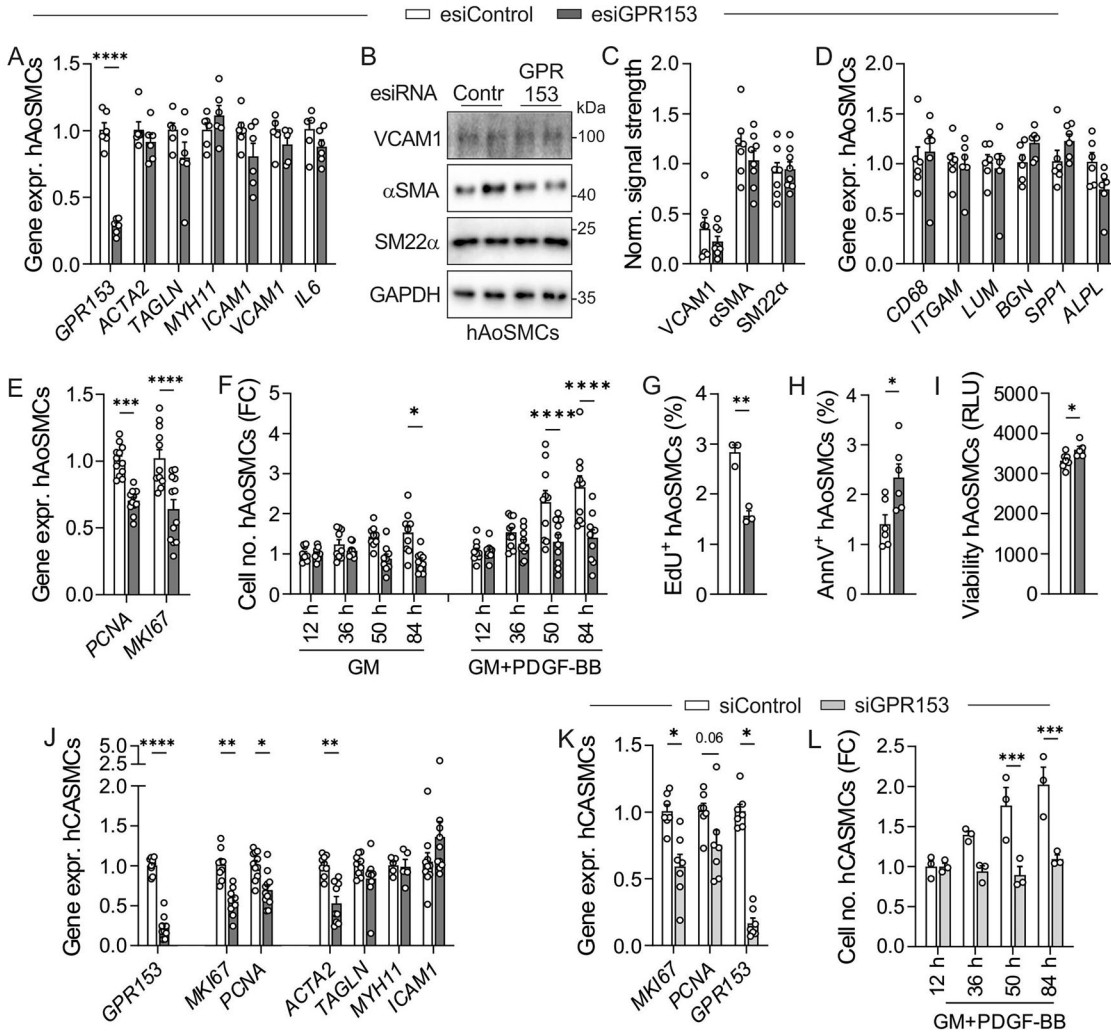

**Fig. 2 | Function of GPR153 in cultured human SMCs. A–E** The expression of marker genes for SMC differentiation was determined in hAoSMCs treated with control esiRNA (esiContr) or GPR153-specific esiRNA (esiGPR153) by qRT-PCR (**A**, **D**, **E**; data normalized to *GAPDH* and control set to 1, $n = 6$–12) or immunoblotting (**B** representative immunoblots; **C** densitometric analysis of signal strength normalized to loading control GAPDH, $n = 7$–8). **F** Cell number changes were determined at the indicated times after seeding of equal numbers of control and GPR153 knockdown hAoSMCs; Cells were cultured in growth medium (GM) with or without 10 ng/ml PDGF-BB ($n = 9$–10). **G** The percentage of EdU-positive cells was determined by flow cytometry 24 h after the addition of 10 μM EdU ($n = 3$, cells cultured in growth medium). **H** The percentage of annexin V-positive (AnnV⁺) cells was determined by flow cytometry ($n = 5$–7). **I** Viability of cells was determined by measuring live-cell protease activity ($n = 5$–7). **J, K** Gene expression was determined by qRT-PCR in hCASMCs after esiRNA-mediated (**J**) or siRNA-mediated (**K**) knockdown of GPR153 (data normalized to *GAPDH*, control set to 1, $n = 7$–10). **L** Cell numbers were determined at the indicated time points after seeding equal numbers of control and GPR153 knockdown hCASMCs ($n = 3$). If not otherwise indicated, experiments were performed after 48 h of serum starvation to induce a differentiated state. Data are means ± SEM; differences between genotypes were analyzed using two-way ANOVA and Sidak's post hoc test (**A**, **C**–**F**, **J**–**L**), unpaired two-sided $t$ test (**G**–**I**). FC fold change; $n$ number of independent knockdown experiments; RLU relative luminescence units; *$P < 0.05$; **$P < 0.01$; ***$P < 0.001$; ****$P < 0.0001$.

(Fig. 2K) was slightly more efficient than esiRNA-mediated knockdown in hAoSMC (Fig. 2A), we chose hCASMC for these analyses. Numerous pathways have been implicated in these processes, but we focused on players known to be regulated by GPCRs, such as ERK1/2, AKT, YAP/TAZ, NF-κB, CREB, or MRTF[15–20]. We studied SMCs after 48 h of serum- and growth factor starvation, which supports a differentiated phenotype, and after exposure to serum- and growth factor-containing medium (growth medium), which induces proliferation and dedifferentiation[31]. In the starved state, GPR153 knockdown did not alter the phosphorylation state of ERK1/2, AKT, or NF-κB p65 (Supplementary Fig. 2A–D), but phosphorylation of YAP was enhanced (Fig. 5A, B). In line with enhanced proteasomal degradation of phosphorylated YAP[32,33], total levels of YAP were reduced (Fig. 5A, C). The same was true for TAZ (Fig. 5A, D), and expression of selected YAP/TAZ target genes was accordingly diminished (Fig. 5E). The

phosphorylation of kinases regulating YAP/TAZ phosphorylation[32,33] was increased in case of LATS1/2, but not clearly changed for MOB1 (Supplementary Fig. 2E–G). In addition to altered YAP/TAZ levels, we found CREB phosphorylation to be significantly elevated (Fig. 5F–I), and also expression of selected CREB target genes[34,35] was increased (Fig. 5J). After stimulation with growth medium, phosphorylation of ERK and AKT was comparable between the genotypes (Fig. 5K–M), whereas NF-κB activation was reduced in GPR153-deficient SMCs (Fig. 5K, N). This was accompanied by reduced expression of selected NF-κB target genes on the mRNA level (Fig. 5O) and the protein level (Supplementary Fig. 2H). Also under these culture conditions, YAP phosphorylation was increased, while total YAP and TAZ levels were reduced (Fig. 5P, Supplementary Fig. 2I–K). Other signaling cascades such as SMAD2/3, STAT1 and KLF4 were not altered in the absence of GPR153, whereas STAT3 phosphorylation was increased

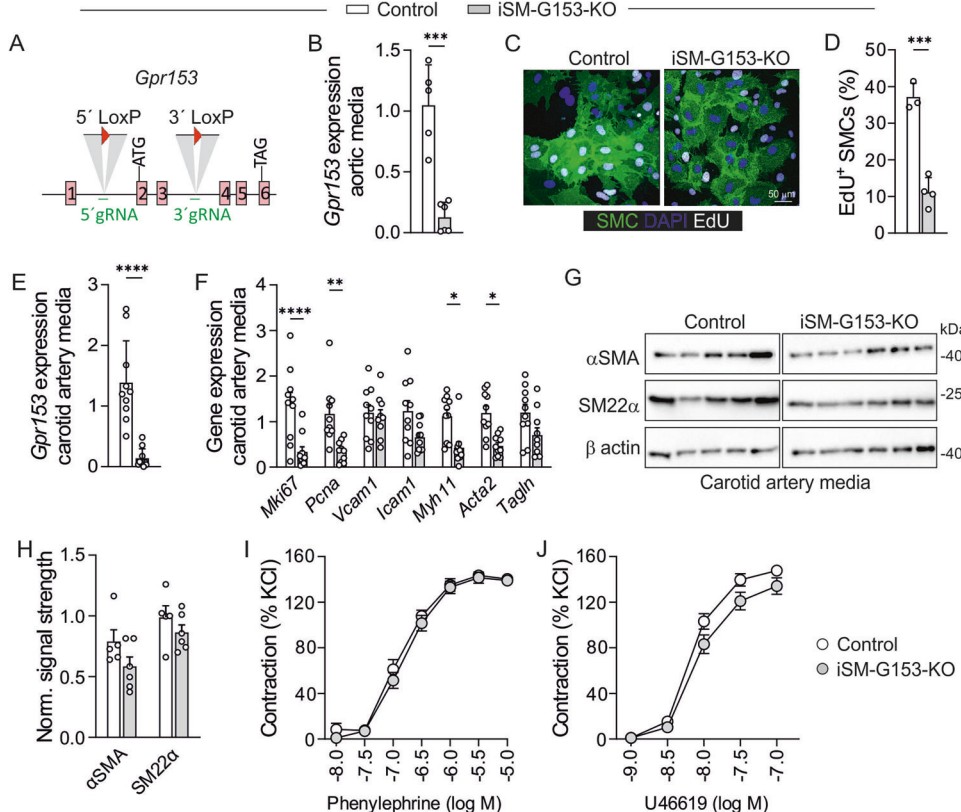

**Fig. 3 | Generation and characterization of tamoxifen-inducible, SMC-specific GPR153 knockout mice (iSM-G153-KOs). A** Design of the floxed *Gpr153* allele. **B** *Gpr153* knockout efficiency was determined by qRT-PCR in the cleaned aortic media of tamoxifen-treated control and iSM-G153-KO mice (data normalized to *Gapdh*, control set to 1, *n* = 5–6). **C, D** EdU incorporation was determined in cells cultured for 2 weeks from digested aortas of control and iSM-G153-KOs (both bred to the mTmG reporter line; SMCs are EGFP-expressing): **C** representative photomicrographs; **D** statistical evaluation (*n* = 3–4). **E, F** qRT-PCR in the cleaned tunica media of carotid arteries from control mice and iSM-G153-KOs (data normalized to *Gapdh* and control set to 1, *n* = 10). **G, H** Immunoblot analysis of carotid arteries (cleaned tunica media) from control mice and iSM-G153-KOs: **G** representative blots; **H** densitometric analysis of signal strength normalized to loading control GAPDH (*n* = 5–6). **I, J** Dose–response curves for the indicated vasoconstrictors were determined by wire myography in mesenteric arteries from control mice and iSM-G153-KOs (data normalized to reference contractions elicited by 60 mM KCl; *n* = 10–14). Data are means ± SEM; comparisons between genotypes were performed using unpaired two-sided *t* test (**B, D, E**), two-way ANOVA and Sidak's multiple comparisons test (**F, H–J**). *n* number of individual mice; *\**P* < 0.05; **\**P* < 0.01; ***\**P* < 0.001; ****\**P* < 0.0001.

(Supplementary Fig. 3A–H). To determine activation of MRTF-A, which is mainly regulated by RhoA-dependent nuclear translocation[15], we performed immunostainings in starved and serum-stimulated hCASMC. Nuclear MRTF-A localization showed a non-significant increase in GPR153-deficient cells in the starved state, whereas no differences were observed in the presence of serum (Fig. 5Q, R). We finally investigated whether corresponding changes were observed in SMCs harvested from aortas of control and iSM-G153-KOs and found that also here, CREB and YAP phosphorylation were enhanced, whereas total TAZ levels were reduced (Fig. 5S–V). Furthermore, increased CREB phosphorylation was observed in carotid arteries harvested from iSM-G153-KOs 7 days after ligation (Fig. 5W, exemplary pictures in Supplementary Fig. 3I). Together, these data show that loss of GPR153 results in SMCs in increased CREB phosphorylation, reduced YAP/TAZ levels, and diminished NF-κB activation with corresponding transcriptional changes in selected target genes.

### GPR153 controls cellular cAMP levels

We next investigated the mechanisms underlying GPR153-mediated regulation of CREB, YAP/TAZ, and NF-κB signaling. The main inducer of CREB phosphorylation is cAMP-dependent PKA activation[36], and cAMP/PKA signaling can negatively regulate both YAP/TAZ and NF-κB[37,38]. $G_s$- and $G_{i/o}$-coupled GPCRs, in turn, are crucial regulators of cellular cAMP levels, which led us to hypothesize that enhanced cAMP levels might underlie both increased CREB phosphorylation and

reduced YAP/TAZ/NF-κB activity in the absence of GPR153. To test this hypothesis, we studied GPR153 effects in human embryonic kidney (HEK) 293 T cells, which allow not only siRNA-mediated knockdown, but also plasmid-based overexpression of GPR153 (Fig. 6A). In addition, plasmid-based reporter systems such as the cAMP GloSensor can be efficiently used in HEK cells. As in SMCs, GPR153 knockdown resulted in HEK cells in enhanced CREB phosphorylation both under basal conditions and after stimulation with $PGE_2$, a known activator of cAMP production via $G_s$-coupled receptors EP2 and EP4 (Fig. 6B, C). Furthermore, GPR153 knockdown was able to enhance CREB phosphorylation induced by forskolin (FSK), a direct activator of adenylyl cyclases (Fig. 6B, C). To determine whether this was associated with altered cAMP levels, we transfected cells with a luminescence-based cAMP sensor and found increased cAMP production after stimulation with FSK or $PGE_2$ in GPR153 knockdown cells (Fig. 6D). Also when using an ELISA-based assay, cAMP production was increased in all tested conditions (Fig. 6E). We next investigated whether overexpression of GPR153 had opposite effects and found that HEK cells transfected with a GPR153 expression plasmid showed reduced cAMP levels both under basal conditions and after stimulation with FSK or $PGE_2$ (Fig. 6F, G). This was accompanied by reduced phosphorylation of CREB (Fig. 6H, I). Importantly, facilitation of cAMP production after knockdown of GPR153 was not only observed in HEK cells, but also in hCASMCs (Fig. 6J, K). To test whether elevated cAMP levels indeed reduced YAP/TAZ and NF-κB signaling, we investigated the effect of

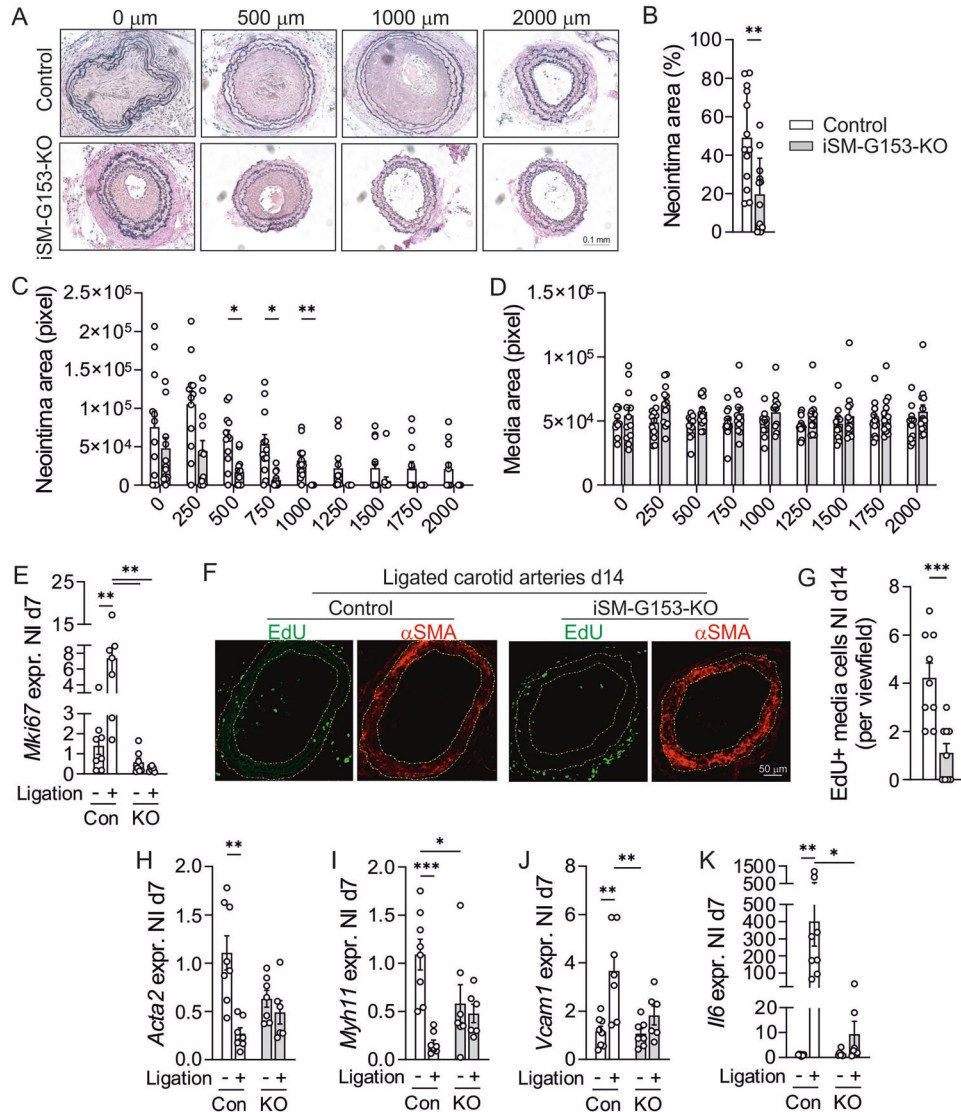

**Fig. 4 | GPR153 regulates SMC proliferation and differentiation in response to vascular damage. A–D** Representative histological sections (**A**) and statistical evaluation of neointima (**B**, **C**) and media (**D**) area at different distances from the ligation site in the carotid artery ligation model (d28). **B** Shows the relative contribution of neointima area to total vessel wall area (average all distances) (n = 12–13). **E, H–K** Expression of select genes was determined by qRT-PCR in the cleaned media of ligated carotid arteries (Ligation+) and non-ligated contralateral vessels (Ligation−) harvested on d7 after induction of neointima (NI) formation from control mice (Con) and iSM-G153-KOs (KO) (data normalized to *Gapdh*, non-ligated control set to 1, n = 6–8). **F, G** Carotid arteries harvested on day 14 after ligation from mice that had received EdU injections every second day from days 6–14 were cryosectioned and stained for αSMA and EdU: Exemplary microphotographs (**F**, hatched yellow line indicates media borders as judged by αSMA stain) and statistical evaluation (**G**, n = 9). Data are means ± SEM; differences between genotypes were analyzed using unpaired two-sided *t* test (**B**, **G**) or two-way ANOVA with Sidak's post hoc test (**C–E**, **H–K**). *n* number of individual mice; *$P < 0.05$; **$P < 0.01$; ***$P < 0.001$.

FSK on TAZ expression and NF-κB activation. We found that FSK-induced CREB phosphorylation was accompanied by reduction in TAZ levels (Fig. 6L, M), and that TNFα/lipopolysaccharide (LPS)-induced NF-κB p65 phosphorylation was diminished by concomitant exposure to FSK (Fig. 6N, O). Also in hCASMC, FSK treatment resulted in increased CREB phosphorylation and consecutively reduced levels of TAZ and NF-κB p65 phosphorylation (Supplementary Fig. 4A–E), and these changes were accompanied by reduced proliferation in control hCASMC, but not in GPR153 knockdown hCASMC (Supplementary Fig. 4F).

As to the mechanism by which GPR153 suppresses cellular cAMP levels, we first investigated whether GPR153 modulates the expression of cAMP synthesizing adenylyl cyclases or cAMP metabolizing phosphodiesterases, but RNA sequencing did not reveal clear differences in GPR153-overexpressing or GPR153 knockdown cells

(Supplementary Fig. 5A). We then determined whether GPR153 has, as described for other orphan GPCRs[39], constitutive activity towards $G_{i/o}$, the G-protein family mediating inhibition of adenylyl cyclases. To address this, we tested whether GPR153-mediated suppression of cAMP was sensitive to $G_{i/o}$ inhibitor pertussis toxin (PTX), which ADP-ribosylates Gα subunits of the $G_{i/o}$ family. Pre-treatment with PTX did not affect GPR153-mediated reduction of cAMP, neither under basal conditions nor in $PGE_2$-stimulated cells (Fig. 6P). This insensitivity towards PTX was not due to inactivity of the toxin, since the same PTX treatment was able to abrogate the inhibitory effect of SDF1α, acting through the $G_{i/o}$-coupled receptor CXCR4, on FSK-induced cAMP production (Supplementary Fig. 5B). Also, prolonged pre-treatment with PTX for 16 h did not affect GPR153-mediated cAMP reduction (Supplementary Fig. 5C). A third hypothesis to explain GPR153-mediated suppression of cAMP levels is an inhibitory effect

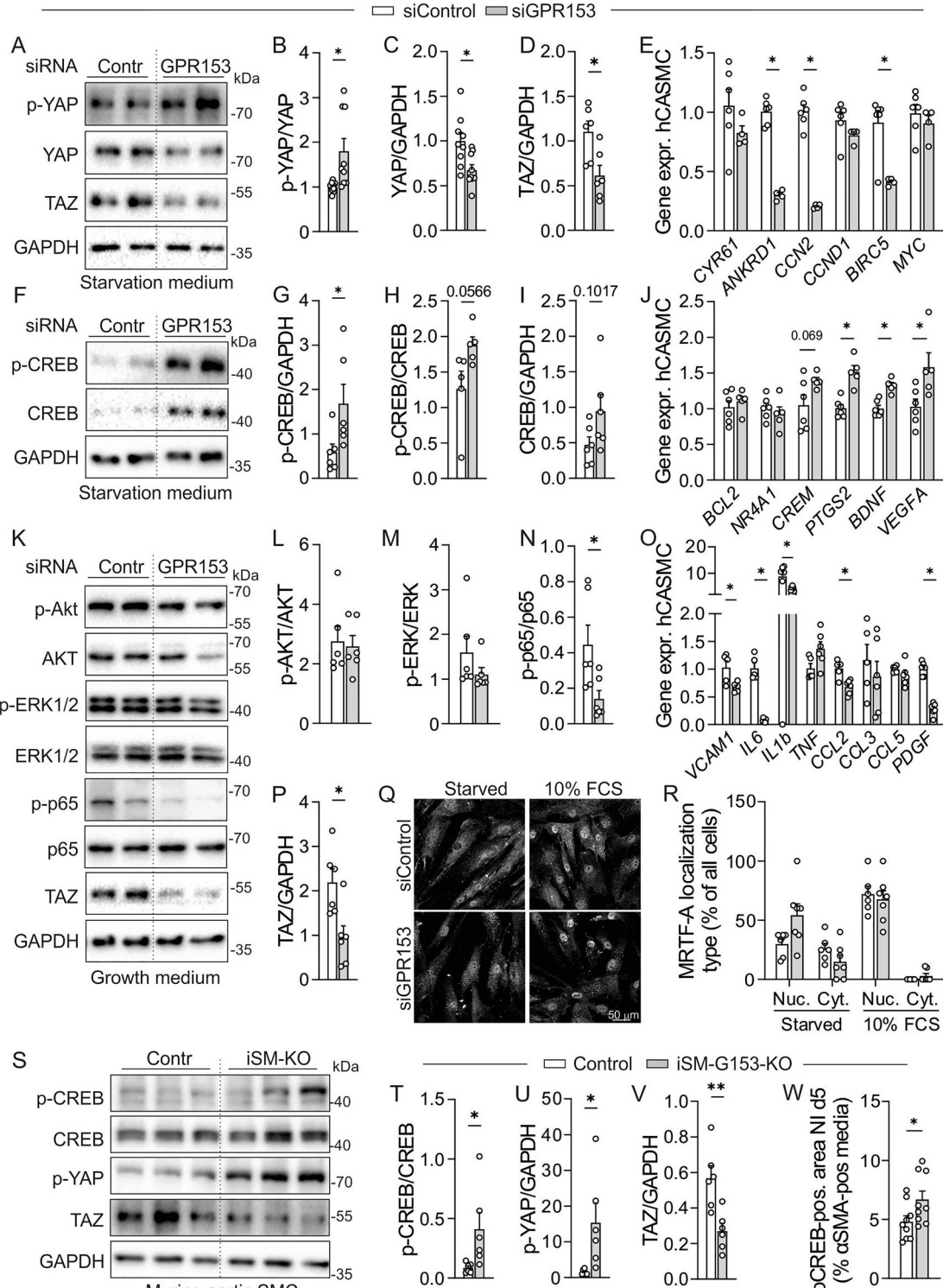

**Fig. 5 | GPR153 regulates CREB, YAP/TAZ, and NF-κB signaling in human coronary artery smooth muscle cells (hCASMCs). A–J** hCASMCs transfected with control siRNA (siControl) or GPR153-specific siRNA (siGPR153) were grown in the absence of serum/growth factors for 48 h, then analyzed by immunoblotting (**A–D, F–I**, GAPDH as loading control, *n* = 5–6) or qRT-PCR (**E, J**; data normalized to *GAPDH*, control set to 1, *n* = 4–6). **K–P** Control and GPR153 knockdown hCASMCs were starved as above and then exposed for 15 min to growth medium (5% FBS, 0.1% insulin, 0.2% basic human fibroblast growth factor, 0.1% human epidermal growth factor), then analyzed by immunoblotting (**K–N, P**; GAPDH as loading control, *n* = 5–6) or qRT-PCR (**O**; data normalized to *GAPDH*, control set to 1, *n* = 4–6). **Q, R** Immunostaining with α-MRTF-A antibodies in serum-starved (48 h) hCASMCs before and after stimulation with 10% FCS for 1 h: representative

photomicrographs (**Q**) and statistical evaluation of the percentage of cells with predominantly nuclear (Nuc.) or mainly cytosolic (Cyt.) MRTF-A localization (*n* = 6-7). **S–V** Immunoblotting of serum-starved aortic SMCs cultured from control and iSM-G153-KOs (GAPDH as loading control, *n* = 6). **W** Carotid arteries harvested on day 7 after ligation from control and iSM-G153-KOs were immunostained with antibodies directed against αSMA and pCREB and pCREB signals within the αSMA-positive area were evaluated (exemplary microphotographs in Supplementary Fig. 3l; *n* = 9). Data are means ± SEM; differences between genotypes were analyzed using unpaired two-sided *t* test (**B–D, G–I, L–N, P, R, T–W**) or unpaired two-sided *t* test with Holm–Sidak's correction for multiple testing (**E, J, O**). *n* number of independent knockdown experiments (**A–R**) or individual mice (**S–W**); *\*P* < 0.05; *\*\*P* < 0.01; *\*\*\*P* < 0.001; *\*\*\*\*P* < 0.0001.

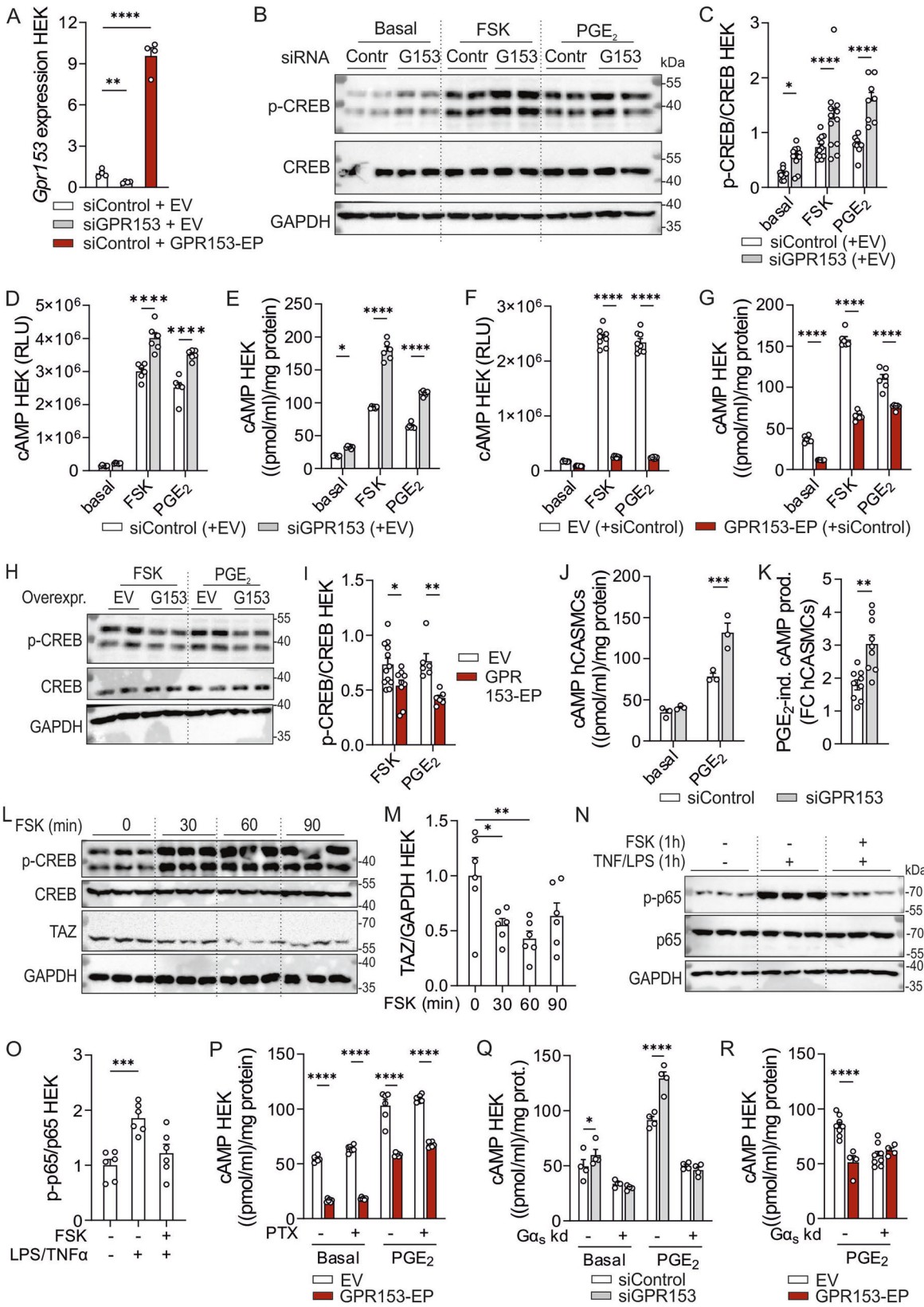

on Gα$_s$-mediated activation of adenylyl cyclases. To test this hypothesis, we investigated whether GPR153-mediated suppression of the cAMP level was preserved after siRNA-mediated knockdown of *GNAS*, the gene encoding Gα$_s$ (Supplementary Fig. 5D). We found that in Gα$_s$-deficient HEK cells, knockdown of GPR153 no longer resulted

in increased cAMP production (Fig. 6Q), and also the suppressive effect of GPR153 overexpression on cAMP production was lost in the absence of Gα$_s$ (Fig. 6R). Taken together, GPR153 reduces cellular cAMP levels through a yet unknown mechanism that is Gα$_{i/o}$-independent, but Gα$_s$-dependent. In the absence of GPR153, cAMP/CREB

**Fig. 6 | GPR153 controls cellular cAMP levels. A** GPR153 expression was determined by qRT-PCR after transfecting HEK cells with control siRNA (siControl) or GPR153-specific siRNAs (siGPR153) as well as empty vector (EV) or GPR153-specific expression plasmid (GPR153-EP) ($n = 4$). **B, C** Immunoblotting of lysates from control and GPR153 knockdown HEK cells in the basal state or after 10 min stimulation with 1 µM FSK or 1 µM PGE$_2$: representative blots (**B**, GAPDH as loading control) and densitometric analysis of signal strength (**C**, $n = 8$–12). **D** cAMP production induced by FSK or PGE$_2$ (each 1 µM, 30 min) was determined in HEK cells transfected with the cAMP GloSensor plasmid as well as control and GPR153-specific siRNAs ($n = 6$). **E** cAMP production induced by FSK or PGE$_2$ was determined in control and GPR153 knockdown HEK cells by ELISA ($n = 6$). **F, G** The effect of GPR153 overexpression on cAMP production was determined in HEK cells using the GloSensor system (**F**, $n = 8$) or ELISA (**G**, $n = 6$). **H, I** Immunoblotting of lysates from EV- and GPR153-EP-transfected HEK cells after 10 min stimulation with 1 µM FSK or PGE$_2$: representative blots (**H**, GAPDH as loading control) and densitometric analysis (**I**) ($n = 6$–12). **J, K** cAMP production induced by PGE$_2$ (1 µM, 30 min) in control and GPR153 knockdown hCASMCs was determined by ELISA: representative experiment (**J**, $n = 3$) and statistical evaluation of PGE2-induced fold changes over basal (FC) in three independent experiments (**K**). **L–O** Effect of 1 µM FSK on total TAZ levels (**L, M**) and p65 NF-κB phosphorylation (**N, O**) ($n = 6$). **P** Effect of PTX (100 ng/ml, 3 h) on PGE$_2$ (1 µM, 30 min)-induced cAMP production was determined by ELISA in EV- and GPR153-EP-transfected HEK cells ($n = 6$). **Q** Effect of Gα$_s$ knockdown on basal and PGE$_2$ (1 µM, 30 min)-induced cAMP production was determined by ELISA in control and GPR153 knockdown HEK cells ($n = 4$). **R** Effect of Gα$_s$ knockdown on PGE$_2$ (1 µM, 30 min)-induced cAMP production was determined by ELISA in EV- and GPR153-EP-transfected HEK cells ($n = 4$–8). Data are means ± SEM; differences between genotypes were analyzed using two-way ANOVA and Sidak's post hoc test (**C–G, I, J, N, P–R**), one-way ANOVA with Dunnett's multiple comparisons test (**M, O**), or unpaired two-sided $t$ test (**A, K**). EV empty vector, FC fold change, FSK forskolin, GPR153-EP expression plasmid encoding GPR153; $n$ number of independent experiments; *$P < 0.05$; **$P < 0.01$; ***$P < 0.001$; ****$P < 0.0001$.

signaling is enhanced, resulting in inhibition of YAP/TAZ- and NF-κB-dependent signals.

## GPR153 regulates CREB, YAP/TAZ, and NF-κB signaling in cultured ECs

Since CREB-, YAP/TAZ-, and NF-κB-dependent signaling also contributes to damage responses in other cell types, we finally investigated whether GPR153 has comparable effects in ECs. Single-cell expression analysis in murine ECs isolated from the vasculature of skeletal muscle or brain showed that the percentage of *Gpr153*-expressing cells is very low in healthy mice, but increases in response to i.p. application of endotoxin LPS (Fig. 7A; data reanalyzed from[23]). Also in human umbilical vein ECs (HUVECs), GPR153 is significantly upregulated by LPS or TNFα, and this effect is blocked by BAY 11−7082, an inhibitor of IκBα phosphorylation and consecutive NF-κB activation (Fig. 7B). siRNA-mediated GPR153 knockdown was efficient in HUVECs (Fig. 7C) and resulted in increased cAMP levels both in the basal state and in response to FSK or PGE$_2$ (Fig. 7D). Furthermore, immunoblotting showed enhanced CREB phosphorylation (Fig. 7E−G) and a reduction in total TAZ (Fig. 7H, I) in GPR153 knockdown cells. In TNFα-stimulated HUVEC, NF-κB p65 phosphorylation was reduced (Fig. 7J, K) upon GPR153 knockdown. To study the consequences of these changes, we determined expression of marker genes for proliferation and inflammatory activation in TNFα-stimulated HUVECs and found reduced expression of adhesion molecules *ICAM1* and *VCAM1* as well as of proliferation markers *PCNA, MKI67*, and *CCNA2* (Fig. 7L). Reduced expression of VCAM1 was confirmed on the protein level (Fig. 7M), whereas proliferation, here determined by EdU incorporation, was not significantly altered (Fig. 7N). To test whether reduced expression of adhesion molecules affected leukocyte adhesion and transmigration, we determined adhesion of human monocytic cell line THP1 and found it to be reduced on TNFα- or LPS-stimulated GPR153-deficient HUVECs (Fig. 7O). Furthermore, chemokine-induced THP1 transmigration through HUVEC monolayers seeded on transwell inserts was reduced after GPR153 knockdown (Fig. 7P). Also the transmigration of primary leukocytes, here freshly isolated human polymorphonuclear granulocytes migrating in response to leukotriene B4, was reduced after GPR153 knockdown in HUVECs (Fig. 7Q). Because elevated cAMP levels are also known to dampen inflammation-induced hyperpermeability in ECs, we investigated the effect of GPR153 deficiency on FITC dextran permeability. GPR153 knockdown HUVECs showed reduced barrier opening induced by proinflammatory mediator thrombin, indicative of improved barrier integrity (Fig. 7R). Taken together, GPR153 knockdown in HUVECs results in increased cAMP/CREB signaling and reduced activity of TAZ and NF-κB pathways. Functional consequences are diminished inflammatory activation in TNFα-stimulated HUVECs, with consecutive reduction of leukocyte adhesion, leukocyte transmigration, and inflammation-induced hyperpermeability.

## In vivo relevance of EC-expressed GPR153

To study the role of endothelial GPR153 in vivo, we focused on brain ECs (BECs), which show upregulation of GPR153 in various models of brain damage (Fig. 8A, data reanalyzed from ref. [40]). To do so, *Gpr153*-flox mice were crossed to *Slco1c1-CreERT2* mice[41], which allow tamoxifen-inducible Cre recombination specifically in ECs of the CNS. QRT-PCR showed a clear reduction of *Gpr153* expression in freshly isolated brain ECs (BECs) from tamoxifen-treated *Slco1c1*-CreERT2; *Gpr153*$^{fl/fl}$ mice (iBEC-G153-KO), but not in lung ECs (Fig. 8B). To investigate the consequences of GPR153 deficiency, we isolated BECs from control and iBEC-G153-KOs and determined gene expression after in vitro expansion and stimulation with TNFα. We found that GPR153-deficient BECs showed impaired expression of adhesion molecules such as *Icam1* and *Vcam1*, and also proliferation marker *Mki67* was reduced (Fig. 8C). Furthermore, transcripts encoding inflammatory mediators such as TNFα, IL-6, or CCL2 were reduced, and diminished CCL2 expression was also observed on the protein level (Fig. 8E). Also lowered ICAM1 and VCAM1 expression was confirmed on the protein level (Fig. 8F), and it was accompanied by reduced adhesion of murine CD4$^+$ T cells and Ly6G$^+$ neutrophils to TNFα-stimulated BECs (Fig. 8G). To investigate the in vivo consequences of reduced inflammatory EC activation, we studied disease susceptibility in experimental autoimmune encephalomyelitis (EAE), a mouse model of multiple sclerosis in which leukocyte recruitment to the CNS depends on EC activation. In this model, first neurological impairment is detected around day 12 after immunization with myelin oligodendrocyte glycoprotein MOG$_{35-55}$, and RNA sequencing in spinal cord ECs harvested on day 12 showed significantly reduced expression of *Vcam1* in GPR153-deficient BECs, and also other inflammatory markers such as cyclooxygenases 1 and 2 (encoded by *Ptgs1* and *Ptgs2*), *Il6*, and *Tnf* were reduced, though not statistically significant (Fig. 8H). In line with reduced inflammatory activity of the blood-brain barrier, we observed reduced immune cell infiltration into the spinal cord on day 14 after immunization (Fig. 8I). To test whether this reduced leukocyte infiltration correlated with reduced neurological impairment, we determined the neurological deficit in controls and iBEC-G153-KOs for four weeks after EAE-induction. These analyses revealed a significantly lower disease severity in iBEC-G153-KOs (Fig. 8J), and histological analysis on day 28 revealed reduced demyelination in iBEC-G153-KOs (Fig. 8K, L).

Since inflammatory EC activation also contributes to the pathogenesis of ischemic brain damage, we subjected mice to permanent distal middle cerebral artery occlusion (MCAO), a mouse model of ischemic stroke[42]. In this model, neutrophils and other immune cells are recruited from the blood stream within hours of stroke onset and can exacerbate tissue damage through the release of reactive oxygen species, proteolytic enzymes, and pro-inflammatory cytokines[42]. To assess the neurological impairment after MCAO, we determined on

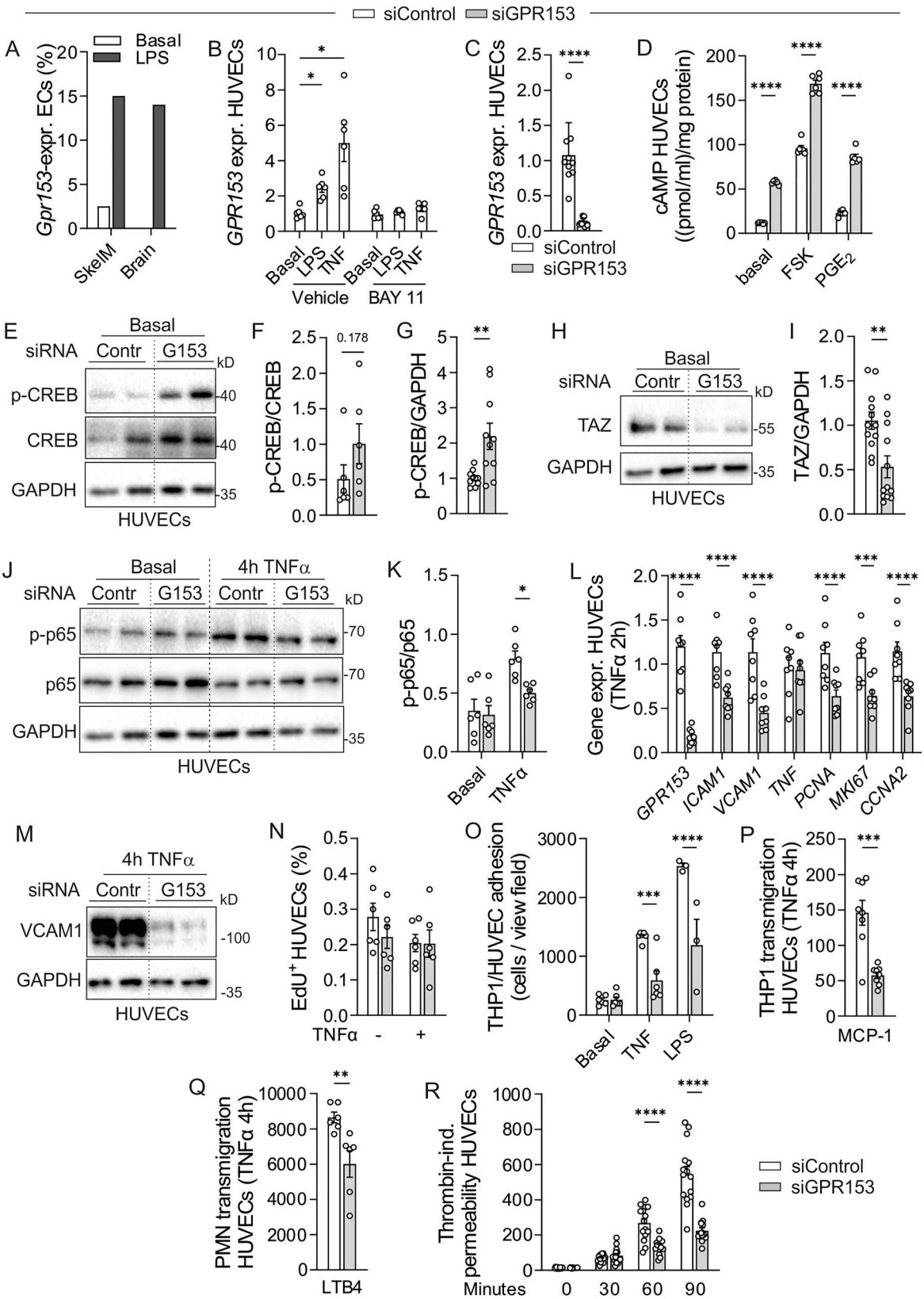

day 2 after surgery the latency to move one body length, a simple motor test of neurological deficit. The latency to move was in both genotypes increased after MCAO, but the effect was less pronounced in iBEC-G153-KOs than in control mice (Fig. 9A). In the corner test, in which after distal MCAO mice are more likely to turn via the contralateral (in this case right) limb[43], KO mice showed less bias towards the right side (Fig. 9B). Also in the tape test, which determines the time until an adhesive sticker attached to the affected forepaw is sensed and removed, KO mice performed significantly better (Fig. 9C). In line with the improved performance in the behavioral tests, infarct volumes were smaller in iBEC-G153-KOs on day 2 after MCAO (Fig. 9D, E), while brain edema was not altered (Fig. 9F). Also 5 days after MCAO, the

**Fig. 7 | GPR153 regulates CREB, YAP/TAZ, and NF-κB signaling in human umbilical vein endothelial cells (HUVECs). A** Percentage of *Gpr153*-expressing ECs isolated from skeletal muscle (skelM) or brain of healthy mice (basal) or 12 h after i.p. injection of LPS 10 mg/kg (data reanalyzed from ref. 23; 117, 34, 52, 22 cells per condition). **B** GPR153 expression was determined by qRT-PCR in HUVECs under basal conditions or 2 h after stimulation with 1 μg/ml LPS or 10 ng/ml TNFα in the presence or absence of BAY 11-7082 ($n = 6$). **C** GPR153 expression in HUVECs treated with control siRNA (siContr) or GPR153-specific siRNA (siGPR153) ($n = 10$). **D** cAMP production in control and GPR153 knockdown HUVECs under basal conditions and after stimulation with FSK or PGE$_2$ (each 1 μM, 30 min) was determined by ELISA ($n = 6$). **E−K** Immunoblot analysis of lysates from control and GPR153 knockdown HUVECs in the basal state or 4 h after stimulation with 10 ng/ml TNFα: representative blots (**E, H, J**; GAPDH as loading control (GAPDH in (**E**) is the same as in (**J**), left)) and densitometric analysis (**F, G, I, K**; $n = 6–12$). **L** Gene expression was determined by qRT-PCR in TNFα-treated control and GPR153 knockdown HUVECs (data normalized to *GAPDH*, control set to 1, $n = 8$). **M** Expression of VCAM1 was determined by immunoblotting in lysates of TNFα (10 ng/ml)-stimulated control and GPR153 knockdown HUVECs (representative of 2 experiments). **N** The percentage of EdU-positive cells was determined by flow cytometry 24 h after the addition of 10 μM EdU ($n = 6$). **O** The number of THP1 cells adhering to control and GPR153 knockdown HUVECs in the basal state or after 6 h of stimulation with 10 ng/ml TNFα or 1 μg/ml LPS was determined by flow cytometry ($n = 3–6$). **P, Q** The number of THP1 cells (**P**) or human blood polymorphonuclear (PMN) leukocytes (**Q**) transmigrating though a monolayer of TNFα (10 ng/ml, 4 h)-treated control and GPR153 knockdown HUVECs was determined in transwell assays with 10 ng/ml MCP1 or 1 μM LTB4 in the lower well ($n = 6–8$). **R** Diffusion of FITC dextran through control or GPR153-deficient HUVEC monolayers was determined before and after stimulation with thrombin (10 units/ml) in a transwell system ($n = 8$). Data are means ± SEM; differences between treatment groups (**B**) or genotypes (**C−R**) were analyzed using unpaired two-sided *t* test (**F, G, I, P, Q**), Mann−Whitney test (**C**), two-way ANOVA and Sidak's post hoc test (**B, D, K, L, N, O, R**). *n* number of independent experiments; *$P < 0.05$; **$P < 0.01$; ***$P < 0.001$; ****$P < 0.0001$.

reduction in infarct volume was significant (Fig. 9G). With respect to mechanism underlying reduced stroke volume in iBEC-G153-KO, histological analyses showed reduced VCAM1 expression in the infarct periphery in iBEC-G153-KOs (Fig. 9H, I), and the number of neutrophils that had extravasated into the brain parenchyma was reduced (Fig. 9J, K).

Taken together, loss of GPR153 leads both in SMCs and ECs to disinhibition of cAMP-dependent signaling, resulting in reduced NF-κB and YAP/TAZ signaling as well increased CREB-dependent gene expression (Fig. 9L). In EC, this results mainly in reduced inflammatory activation and consecutively reduced immune cell recruitment in models of neuroinflammation, whereas in SMC, reduced proliferation and dedifferentiation protected mice from adverse vascular remodeling in models of neointima formation (Fig. 9L).

## Discussion

Endothelial activation and smooth muscle dedifferentiation are important steps in the vascular response to inflammatory, mechanical, or hypoxic damage. We show here that inactivation of GPR153 in SMCs reduces proliferation, alters differentiation, and causes partial protection from neointima formation. In the endothelium, GPR153 deficiency reduces inflammatory activation and consecutive leukocyte adhesion, resulting in vivo in reduced severity of neuroinflammation and ischemic damage. With respect to the underlying mechanisms, we observed increased cAMP levels in GPR153-deficient SMCs, ECs, and HEK cells, and phosphorylation of cAMP effector CREB was consecutively enhanced. Conversely, overexpression of GPR153 reduced in HEK cells cAMP production and CREB phosphorylation.

The cAMP/PKA/CREB pathway is a well-known player in the regulation of vascular cell function. In SMCs, CREB levels are high in differentiated vascular SMCs, but decrease during phenotypic modulation[44], and numerous studies suggested that this pathway mediates anti-mitogenic effects[20]. For example, facilitation of cAMP/PKA/CREB-dependent signaling by knockout of phosphodiesterase PDE3A reduced SMC proliferation[45], and the same was true for overexpression of constitutively active CREB mutants[44–46]. Consistent with this, CREB depletion or inhibition by siRNA or dominant-negative mutants was sufficient to induce SMC proliferation[44,46,47]. However, some studies also indicated that CREB can mediate pro-mitogenic effects, for example those of angiotensin II or thrombin[48,49], and it was suggested that the mode of CREB activation is decisive for its net effect on SMC proliferation[50]. Our data clearly support the notion that elevated cAMP/CREB signaling is associated with reduction of both proliferation and dedifferentiation in human and murine SMCs. Interestingly, we also observed mild enhancement of vascular relaxation in response to G$_s$-coupled receptor agonist cicaprost, which might be due to increased cAMP production in GPR153-deficient SMCs.

In EC, cAMP signaling is known to regulate inflammatory activation and permeability[3,51]. The cAMP effector Epac1, for example, stabilizes the endothelial barrier through activation of Rap1, thereby reducing vascular leakage in response to proinflammatory stimuli[3]. In line with this, elevated cAMP levels in GPR153 knockdown HUVECs were associated with reduced agonist-induced barrier opening. In addition, cAMP-dependent activation of CREB has been shown to interfere with p65-NF-κB activation[38,52], and we confirmed in HEK cells that elevation of the cAMP production by FSK reduces LPS/TNFα-induced p65 phosphorylation. In GPR153-deficient ECs, elevated cAMP levels are accompanied by reduced p65-NF-κB phosphorylation, reduced expression of inflammatory mediators, and diminished leukocyte adhesion and transmigration. We postulate that these changes are the main factors underlying the partial protection of iBEC-G153-KOs in EAE and ischemic stroke. In contrast to SMCs, GPR153-deficient ECs did not show reduced proliferation, even though expression of proliferation markers was diminished on the mRNA level. The reasons for these differences are currently unknown, but EC-specific regulation of protein stability or posttranslational modification might be contributing factors.

Reduced NF-κB activity in the absence of GPR153 was also observed in dedifferentiating SMCs, and we hypothesize that these changes contribute to reduced neointima formation in iSM-G153-KOs. Various studies provided evidence for a role of NF-κB in phenotypic switching, for example, the finding that SMC-specific repression of NF-κB signaling by a truncated IκB attenuated SMC dedifferentiation and neointima formation following vascular injury[17]. Furthermore, SMC-specific ablation of NEMO, an IKK complex subunit that is essential for canonical NF-κB activation, resulted in reduced proliferation, migration and inflammatory gene expression in SMCs, and atherosclerosis development was diminished[53]. In line with this, upregulation of typical NF-κB target genes such as *Vcam1* or *Il6* was reduced in SMCs harvested from ligated carotid arteries of iSM-G153-KOs. It is also possible that altered NF-κB activation contributed to the observed preservation of contractile gene expression, since NF-κB was shown to repress myocardin expression in concert with KLF4[17].

The third pathway that is altered in GPR153-deficient vascular cells is YAP/TAZ signaling. YAP and TAZ shuttle between cytoplasm and nucleus and regulate cell functions such as proliferation, differentiation, and survival by acting as coactivators for TEAD family transcription factors[33,54,55]. In SMCs, YAP/TAZ were suggested to facilitate dedifferentiation, since overexpression of YAP enhances SMC proliferation and migration, whereas lentiviral knockdown reduces dedifferentiation in vitro and in vivo[56,57]. Based on these data it is tempting to speculate that reduced YAP/TAZ levels contribute to impaired proliferation and dedifferentiation in GPR153-deficient SMCs, but the relative contribution compared to altered CREB and NF-κB phosphorylation is unclear. It should also be noted that the role of YAP/TAZ

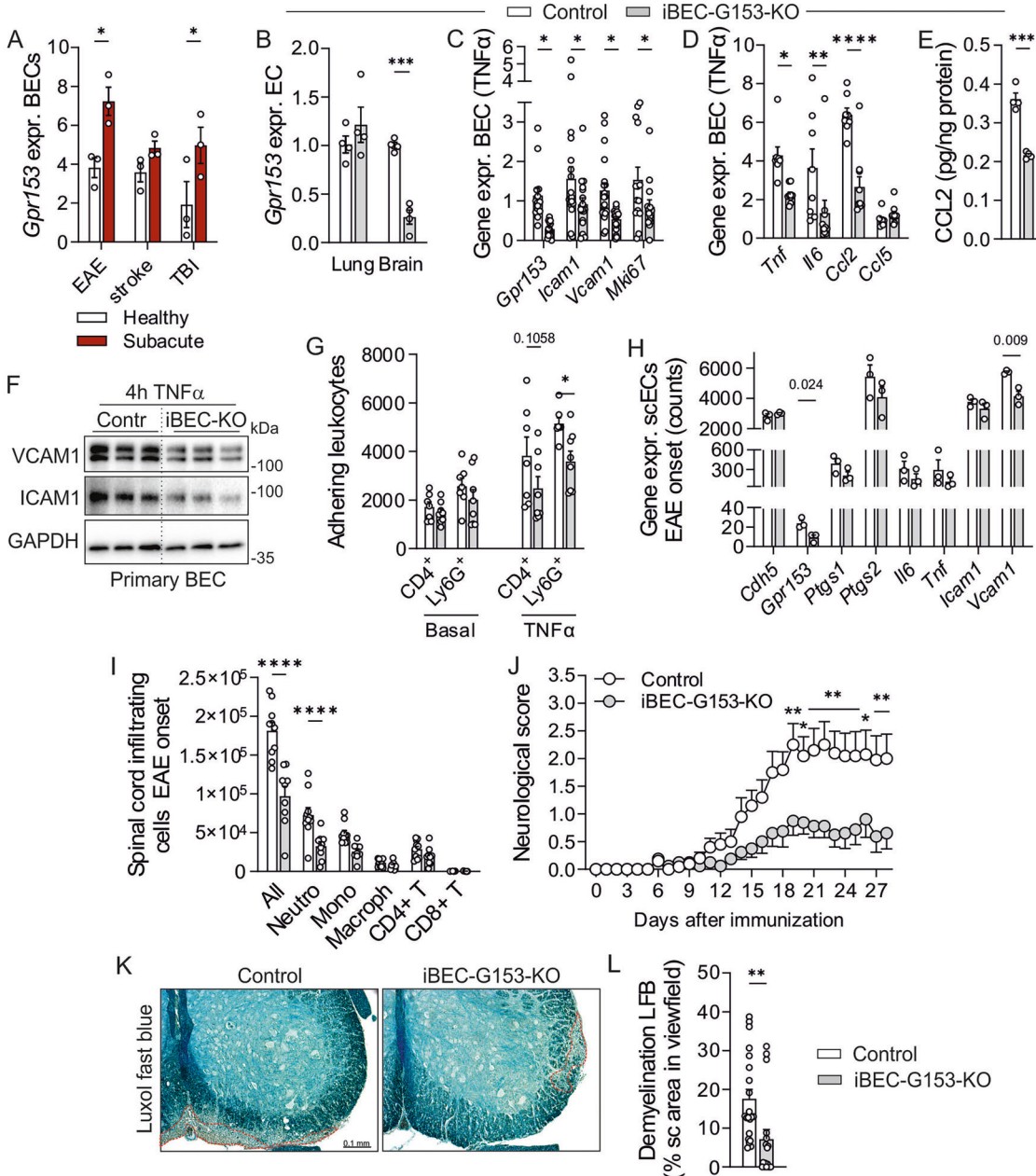

**Fig. 8 | Generation and analysis of mice with brain EC-specific GPR153 deficiency (iBEC-G153-KOs). A** *Gpr153* expression was determined by RNAseq in BECs from mice subjected to different forms of CNS injury (data reanalyzed from ref. 40; $n = 3$). **B** GPR153 expression was determined in CD31[+],CD45[−] cells sorted by flow cytometry from lungs and brains of tamoxifen-treated control mice and iBEC-G153-KOs ($n = 4$). **C, D** Gene expression was determined by qRT-PCR in TNFα (10 ng/ml, 2 h)-stimulated BECs cultured from control mice and iBEC-G153-KOs (data normalized to *Gapdh*, control set to 1, $n = 9$–16). **E** CCL2 levels in the supernatant of TNFα (10 ng/ml, 4 h)-stimulated BECs from control mice and iBEC-G153-KOs were determined by ELISA ($n = 4$). **F** Expression of ICAM1 and VCAM1 was determined by immunoblotting in lysates of TNFα (10 ng/ml)-stimulated BECs from control and iBEC-G153-KO mice (representative of 2 experiments). **G** Adhesion of mouse blood CD4[+] T cells and Ly6g[+] neutrophils to basal and TNFα (10 ng/ml, 4 h)-stimulated BECs grown from control and iBEC-G153-KOs was determined by flow cytometry ($n = 3$–6). **H**–**L** Experimental autoimmune encephalomyelitis (EAE): **H** mRNA

sequencing in CD31[+],CD45[−] ECs sorted by flow cytometry from spinal cords (scECs) on day 12 after induction of EAE by MOG$_{35-55}$ immunization (onset of symptoms, $n = 3$). **I** Flow cytometric analysis of spinal cord-infiltrating leukocytes on day 14: Absolute numbers of CD45[+] leukocytes (All), CD45[+],CD11b[+],Ly6G[+] neutrophils (Neutro), CD45[+],CD11b[+],Ly6G[−],Ly6C[hi] monocytes (Mono), CD11b[+],Ly6G[−],CD45[hi],Ly6C[−] macrophages (Macroph), CD45[+],CD19[−],CD4[+] T cells (CD4 T), CD45[+],CD19[−],CD8[+] T cells (CD8 T) ($n = 9$–10). **J** Neurological score after MOG$_{35-55}$ immunization in control mice and iBEC-G153-KOs ($n = 8$–10). **K, L** Quantification of demyelination by luxol fast blue staining on day 28 after induction of EAE: exemplary photomicrographs (**K**) and statistical evaluation (**L**) (18 and 21 sections from 6 and 7 mice, respectively). Data are means ± SEM; comparisons between genotypes were performed using two-way ANOVA with Sidak's multiple comparison test (**A**–**D, G, I, J**), unpaired two-sided *t* test (**E, L**), unpaired two-sided *t* test with Holm–Sidak's correction for multiple testing (**H**). *n* number of individual mice; *$P < 0.05$; **$P < 0.01$; ***$P < 0.001$.

in SMCs might be more complex than previously acknowledged, since recent studies showed increased dedifferentiation and osteogenic transdifferentiation in SM-specific YAP/TAZ-KOs[58,59]. In EC, YAP/TAZ have been shown to contribute to inflammatory activation[60], and

reduced YAP/TAZ levels in GPR153-deficient ECs might contribute to diminished leukocyte adhesion. However, the role of YAP/TAZ in EC inflammation was mainly studied in the context of disturbed flow, whereas our studies were performed under static conditions. It is

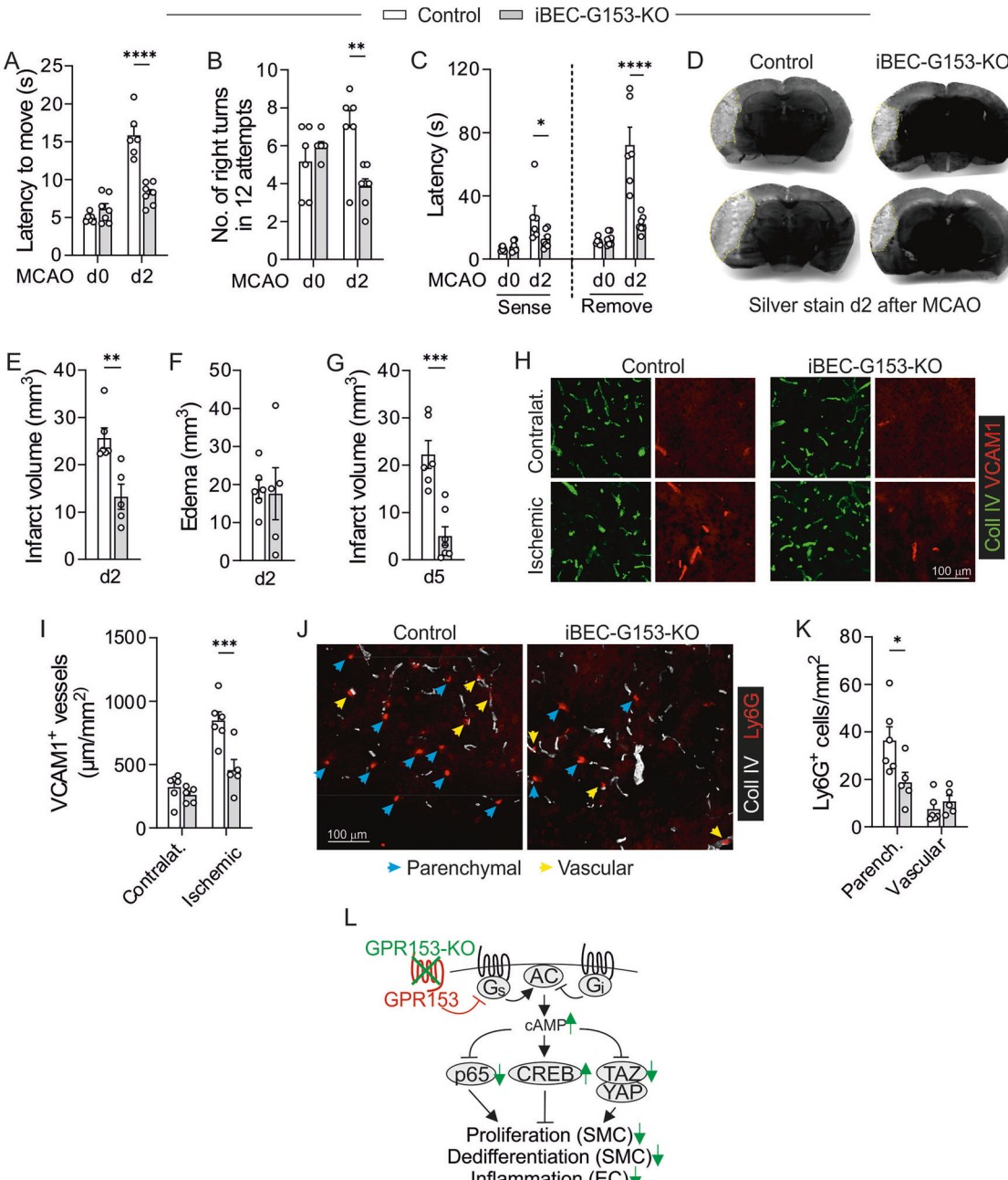

**Fig. 9 | In vivo relevance of EC-expressed GPR153: Medial cerebral artery occlusion (MCAO).** **A–C** Behavioral testing before (d0) and 2 days (d2) after MCAO: latency to move one body length (**A**), number of right turns in the corner test (**B**), and latency to sense and remove an adhesive sticker in the tape test (**C**) ($n = 6$–7). **D–F** Histological analysis of brains harvested on d2 after MCAO: representative photomicrographs of silver-stained brain sections (**D**, hatched yellow lines indicate infarct border), statistical evaluation of infarct volume (**E**) and edema volume (**F**) ($n = 5$–6). **G** Infarct volume day 5 (d5) ($n = 6$–7, same mice as in (**A–C**)). **H, I** Immunofluorescence staining of brain sections (same individuals as in (**D–F**)) with antibodies directed against collagen IV (Coll IV, stains vascular basement membrane) and VCAM1: representative photomicrographs (**H**) and statistical

evaluation of the cumulative length of VCAM1-positive vessels (**I**, $n = 5$–6). **J, K** Immunofluorescence staining with antibodies directed against Coll IV and neutrophil marker Ly6G: representative photomicrographs (**J**) and statistical evaluation of Ly6G-positive cells outside (parenchymal, blue arrows) or in direct contact with Coll IV-stained vessels (vascular, yellow arrows) (**K**, $n = 5$–6). **L** Graphical summary of the proposed role of GPR153; changes in GPR153-deficient ECs and SMCs are shown in green; for details, see text. Data are means ± SEM; comparisons between genotypes were performed using two-way ANOVA with Sidak's multiple comparison test (**A–C, I, K**), unpaired two-sided $t$ test (**E–G**). $n$ number of individual mice; *$P < 0.05$; **$P < 0.01$; ***$P < 0.001$.

therefore unclear to what degree reduced YAP/TAZ levels contribute to impaired inflammatory gene expression in GPR153-deficient ECs.

As to the question of how GPR153 regulates YAP/TAZ levels, we postulate that also here elevated cAMP levels are causal. Previous studies suggested that elevation of cAMP using FSK or agonists at $G_s$-coupled receptors enhances phosphorylation, nuclear export, and degradation of YAP/TAZ, resulting in reduced TEAD-luciferase

reporter gene activity[37]. We confirmed that activation of cAMP signaling by FSK reduces TAZ levels, and we hypothesize that the same applies for cAMP elevation induced by loss of GPR153. Mechanistically, GPCRs were suggested to regulate YAP/TAZ phosphorylation through regulation of the Hippo pathway, in which kinases MST1/2 phosphorylate and thereby activate LATS1/2, which in turn phosphorylate YAP and TAZ, thereby promoting their retention and degradation in the

cytosol[32,33]. In accordance with this, we found increased LATS1/2 phosphorylation in GPR153-deficient SMCs, indicating that increased activity of the Hippo pathway at least partially underlies the observed changes in YAP/TAZ signaling.

The mechanism by which GPR153 suppresses cAMP levels is currently not understood and probably differs from canonical G-protein signaling. A previous study showed that certain orphan GPCRs have constitutive activity towards $G\alpha_{i/o}$ family G-proteins, resulting in inhibition of cAMP-mediated effects[39]. GPR153 was not analyzed in that particular study, so we tested whether GPR153-mediated reduction of cAMP was sensitive to the $G_{i/o}$ inhibitor PTX, but found this not to be the case. Also, other studies did not find evidence for constitutive activity of GPR153 towards $G\alpha_{i1}$ or other G-proteins[61,62]. Instead, we found that GPR153-mediated suppression of cAMP levels depended on the presence of $G\alpha_s$, suggesting that GPR153 acts by interfering with $G\alpha_s$-dependent adenylyl cyclase activation. Whether this is achieved via direct interaction between GPR153 and $G\alpha_s$ or rather through interaction of GPR153 with $G_s$-coupled receptors, is currently unclear. Dimerization between GPCRs has been suggested to modulate signaling efficiency and G-protein coupling[63,64], but potential dimerization partners of GPR153 are not known. However, the fact that cAMP levels are not only altered in response to defined GPCR ligands such as $PGE_2$, but also under basal conditions and after stimulation with FSK, indicates that GPR153 either interacts with several $G_s$-coupled receptors or indeed directly modulates $G\alpha_s$-dependent adenylyl cyclase activation.

In summary, our data show that the expression level of GPR153 has a significant impact on cellular cAMP levels, which raises the question of how GPR153 expression is regulated. Our transcriptional analyses indicated that endothelial GPR153 expression is under basal conditions very low, but that it is both in human and murine ECs upregulated in response to inflammatory stimuli in an NF-κB-dependent manner. Also in SMCs, GPR153 is upregulated in response to mechanical, metabolic, or ischemic damage. Since GPR153 reduces cAMP production and thereby dampens cAMP/PKA/CREB-mediated anti-inflammatory processes, upregulation of GPR153 might represent a positive feedback loop facilitating inflammation and dedifferentiation. Whether GPR153-dependent regulation of cAMP levels is ligand-dependent, is currently unclear. So far, no ligands for GPR153 have been identified, and our data show that cAMP levels are altered in the absence of exogenous stimuli. However, it is possible that yet unknown SMC- or EC-autonomous ligands play a role.

Taken together, our data show that GPR153 facilitates inflammation and dedifferentiation both in ECs and SMCs, and that genetic inactivation of this receptor ameliorates neuroinflammation and vascular remodeling. In the absence of GPR153, inflammatory activation is reduced both in ECs and SMCs, and this includes the reduced production of pro-inflammatory soluble mediators, which in turn might affect the activation state of neighboring cells. Importantly, the modulatory effect of GPR153 on vascular cells is not restricted to murine cells, but is also observed in primary human ECs and SMCs. These findings, together with the fact that the receptor is strongly upregulated in response to damage, make GPR153 an interesting target for parallel modulation of EC inflammation and SMC dedifferentiation. However, our findings are currently limited to the murine models of ligation-induced neointima formation, EAE, and ischemic stroke, and further studies are required to determine the relevance of GPR153 in other vascular damage models.

## Methods

### Ethical approval

All animal experiments were approved by the Institutional Animal Care and Use Committee of the *Regierungspräsidium* Darmstadt and in accordance with Directive 2010/63/EU of the European Parliament on the protection of animals used for scientific purposes. Analyses were performed by investigators blinded to genotype and treatment.

### Reagents

Recombinant human TNF-α (#300-01 A) was purchased from Peprotech, leukotriene B4 (#Cay20110) from Cayman, human CCL2/MCP-1 (#279-MC) from R&D Systems. Tamoxifen (#T5648) and lipopolysaccharide (#L4391) were purchased from Sigma-Aldrich.

### Experimental animals

Mice carrying a floxed *Gpr153* allele were generated by CRISPR/Cas9 genome editing in murine zygotes. crRNAs targeting GPR153 intron 1–2 (upstream guide, 5′- GGTACACACCCTTAACGGTC-AGG -3′) and GPR153 intron 3–4 (downstream guide, 5′- GAGGCGACTCAATAGTCAAA-GGG -3′) were designed using https://eu.idtdna.com/site/order/designtool/index/CRISPR_CUSTOM. Homology-directed repair DNAs were designed by flanking the 34 base pair (bp) loxP sequence with 60 bp homology arms corresponding to intronic sequences flanking the PAM site: Repair DNA for the upstream guide was 5′-TCTGACGCCCTCTTCTGGGCTCT GAAGGTATCTGCATGCATGTGGTACACACCCTTAACG**ATAACTTCGTA TAATGTATGCTATACGAAGTTAT**GTCAGGCCCACTGTGGCTGGAGAG ACTGCTCAGTGGTTAAGAGCACCAACTACTTACTCT-3′ (bold: loxP sequence), Repair DNA for the downstream guide was 5′-TTTGCAAA-CAGATGTGGGGACCCTGGTAAGAGCCCTAAAGGCCATCTGGTTAGACC CTTT**ATAACTTCGTATAATGTATGCTATACGAAGTTAT**GACTATTGAG TCGCCTCTAGGGTCTGAGCTTCTGGGTAGGGTCAGAACTGTGGCCCAG GA-3′ (bold: loxP sequence). For genome editing, the Alt-R CRISPR-Cas9 system (Integrated DNA Technologies) was applied. Briefly, the self-designed crRNAs were annealed with tracrRNA (1072532; IDT) to form the complete guide RNA. A mixture of 100 ng/µl Cas9 nuclease (1081058; IDT), 200 ng/µl of each complete guide RNA, and 115 ng/µl of each HDR DNA was electroporated into pronuclei of C57BL/6 J zygotes during the 2-cell stage as described previously[65]. This resulted in the correct insertion of the upstream loxP site in two male mice, and their offspring were used to target selectively the downstream loxP site in a second round of zygote electroporation. Correct integration of both loxP sites was confirmed by DNA sequencing in one founder female, which was backcrossed to C57BL/6 J to give rise to the *Gpr153*-flox line. Genotyping for *Gpr153* was done using primers 5′-TTCAGTTCCCAG-CATCCCTC-3′ and 5′-ATGTGATGTGAGCCACCATG-3′, resulting in band sizes of 222 bp for the wild type allele and 256 bp for the floxed allele.

To generate mice with tamoxifen-inducible, SMC-specific GPR153 deficiency (iSM-G153-KO), *Gprc153*-flox mice were intercrossed with *Myh11*-CreERT2 mice[30]. Mice with tamoxifen-inducible recombination in brain ECs (iBEC-G153-KO) were generated using *Slco1c1*-CreERT2[41]. Mice were maintained on a C57BL/6 J background, and genetically matched Cre-negative *Gpr153*^fl/fl^ mice were used as controls. To allow flow cytometric isolation of vascular SMCs, some mice carried in addition the Cre-dependent fluorescent reporter construct mTmG (Rosa26flox-mT-stop-flox-mG (Jackson Lab, Stock 007576))[19]. In these cases, genetically matched *Myh11*-CreERT2-positive; *Gpr153*^wt/wt^; mTmG^fl/fl^ mice served as controls for *Myh11*-CreERT2-positive; *Gpr153*^fl/fl^; mTmG^fl/fl^ mice.

For induction of Cre-mediated recombination, mice were treated with 1 mg tamoxifen (Sigma, T5648) dissolved in Miglyol 812 (3274, Caelo) intraperitoneally for five consecutive days. If not otherwise indicated, experiments were performed 4 to 12 days after the end of tamoxifen induction. Mice were kept in individually ventilated microisolator cages with 12-h light-dark cycles in a specific pathogen-free facility with a temperatures range between 20–24 °C and humidity between 45–65%. Standard chow (Altromin 1320, Altromin GmbH Lage), tap water, and nesting material were provided ad libitum.

### Disease models

Mice were analyzed at an age of 7–16 weeks unless stated otherwise. Since the *Myh11*CreERT2 BAC transgene has integrated on the Y chromosome, only male mice were used in these studies. Studies in

iBEC-G153-KOs were performed in female mice (EAE) or both genders (MCAO, cell culture, transcriptional analyses).

**For induction of neointima formation**, the carotid artery ligation model was used. In brief, the left common carotid artery was ligated using a 6-0 silk suture in anesthesia with ketamine (120 mg/kg) and xylazin (16 mg/kg) with intra- and postoperative analgesia with metamizole (0,8 ml/500 ml drinking water) and buprenorphine (0.05–0.1 mg/ml s.c.). After 4 weeks, mice were sacrificed with $CO_2$ and carotid arteries were surgically excised. To detect EdU incorporation in carotid arteries, a subgroup of mice received subcutaneous injections of 30 mg/mL EdU (50 μL per injection) on days 6, 8, 10, 12, and 14 after ligation, and vessels were harvested on day 14. **To induce EAE**, female mice were immunized subcutaneously with MOG$_{35-55}$ /CFA emulsion (EK-2110, Hookelabs). On days 0 and 1, mice received 500 ng pertussis toxin (EK-2110, hookelabs) intraperitoneally. From day 7 on, animals were monitored daily until day 28, and clinical scoring of EAE was conducted as follows: 0, no clinical disease; 1, limp tail; 2, impaired righting reflex and gait; 3, partial hind limb paralysis; 4, tetraparesis; 5, dead.

As a model of ischemic stroke, **distal permanent middle cerebral artery occlusion (MCAO)** was performed. At an age of 8–16 weeks, age- and sex-matched mice of both genotypes were treated with tamoxifen (1 mg i.p., dissolved in 90% miglyol 812/10% ethanol) twice per day for five days to induce recombination. Two weeks after the first tamoxifen injection, behavioral tests were performed (see below), then mice were anaesthetized by ketamine (65 mg/kg body weight) and xylazine (14 mg/kg). During surgery, mice were kept on a heating pad, and body temperature was controlled using a rectal probe. Under a microscope, a skin incision was made between the ear and the orbit on the left side. After removing the temporal muscle and drilling a burr hole, the middle cerebral artery was occluded by microbipolar electrocoagulation (Model ICC, Erbe). Two days after MCAO, behavioral tests were repeated. Two or five days after MCAO, mice were deeply re-anaesthetized and transcardially perfused with Ringer solution containing heparin (10 IU/ml). To evaluate neurological function before and after MCAO, we used three behavioral tests. In the corner test, mice were allowed to enter a 30 × 20-cm corner with an angle of 30°, and the number of right and left turns on rearing out of 12 trials was counted. For the latency-to-move test, mice were placed at the center of a plain board and the time to cross one body length (7 cm) was measured. In the sticky-tape-removal test, a small circular adhesive tape (HERMA No 2212, 8 mm) was put onto forepaws, and the time when mice first tried to remove the tape (sensation time) as well as the total time needed to remove it (removal time) were determined.

**For telemetric blood pressure measurements** in conscious, unrestrained mice, a radiotelemetry system (Data Sciences International, PA-C10) was inserted into the left common carotid artery in anesthesia with ketamine (120 mg/kg) and xylazin (16 mg/kg) with intra- and postoperative analgesia with metamizole (0,8 ml/500 ml drinking water) and buprenorphine (0.05–0.1 mg/ml s.c.). For angiotensin II-induced arterial hypertension, osmotic pumps (Alzet, model 2004) releasing angiotensin II (2000 ng/kg/min) for 28 days were implanted subcutaneously under anesthesia with isoflurane and intra- and postoperative analgesia with metamizole (via drinking water) and local infiltration with bupivacaine (1 mg/kg). Data were acquired and analyzed using Ponemah software (Data Sciences International).

## Culture conditions for cell lines
**Human aortic SMCs** (hAoSMCs; CC-2571) and **human coronary artery SMCs** (hCASMCs; CC-2583) were purchased from Lonza (male and female donors) and cultured in SMC basal medium (SmBM, CC-3181) supplemented with 5% FBS, 0.1% insulin, 0.2% basic human fibroblast growth factor (hFGF-b), 0.1% GA-1000, and 0.1% human epidermal growth factor (hEGF) (SmGM2; CC-4149) (growth medium). All SMCs were used for a maximal number of 15 passages. If not otherwise indicated, analyses were performed after inducing a differentiated SMC phenotype by 48 h of culture in growth factor-reduced medium (SmBM with 0.1% FBS) (starvation medium).

**Human umbilical vein ECs** (HUVECs; CC-2519) were purchased from Lonza (male and female donors) and cultured in EC Growth Basal Medium-2 (EBM-2; Lonza CC-3156) containing the EGM2 supplement (Lonza CC-4176: 2% FBS, 0.04% hydrocortisone, 0.4% hFGF-B, 0.1% VEGF, 0.1% R3-IGF-1, 0.1% ascorbic acid, 0.1% hEGF, 0.1% GA-1000 and 0.1% heparin) (=EGM2 medium). HUVECs were used for a maximal number of 8 passages and cultured on polystyrene plates (Greiner bio-one) coated with 30 μg/ml collagen type I (354236; Corning). In some cases, HUVEC were pretreated for 1 h with 1 μM BAY 11-7082 (B5556, Sigma-Aldrich).

**Human embryonic kidney cells** (HEK293T, ATCC) cells were maintained in DMEM (10938025, Thermo Scientific) supplemented with 10% FBS (10270106, Thermo Scientific), 1% penicillin and streptomycin (15140122, Thermo Scientific), 1% L-glutamine (25030024, Thermo Scientific) and 1% sodium pyruvate (11360070, Thermo Scientific). All polystyrene plates for HEK293T cell culture were coated with poly-L-lysine hydrobromide (P9155; Sigma-Aldrich) for at least 2 h.

**Human leukemic monocyte cell line THP1** was obtained from Sigma-Aldrich (88081201) and cultured in RPMI 1640 Medium (21875034; Thermo Scientific) supplemented with 10% FBS, penicillin and streptomycin, 2mM L-glutamine, and 50uM 2-Mercaptoethanol (31350010; Thermo Scientific). As floating cells, THP1 cells were maintained in flasks at a cell density of 3–5 × 100,000 cell/ml.

All cells were cultured at 37 °C and 5% $CO_2$ and were tested negative for mycoplasma contamination (MP0035; Sigma-Aldrich) before experiments.

## Primary cell isolation and culture
**Murine primary aortic SMCs** were isolated from male mice as previously described[66]. Briefly, freshly extracted aortas were cleaned of surrounding tissue and adventitia. The aortas were then cut open longitudinally, and ECs were scraped off using a forceps. Samples were cut into small pieces with a scalpel and digested with 5 mg/ml collagenase II (4174, Worthington Biochemical Corporation) in culture medium (2 ml DMEM, 20% FBS, 1% PenStrep) at 37 °C for 30 min while shaking at 700 rpm. The resulting SMCs were further homogenized by pipetting and then further processed as follows: **For direct cell culture**, homogenized cells were centrifuged at 500 x *g* for 5 min, then cell pellets were resuspended in culture medium and seeded into one well of a 0.2% gelatin (G1393; Sigma-Aldrich)-coated 6-well plate (cells from one aorta per well). Cells were cultured undisturbed for 3 to 5 days until fully adherent. After that, medium was changed every second day until cells reached confluency around day 14. **For sorting of SMCs from mTmG reporter mice**, homogenized SMCs were filtered through a 100-micron strainer and centrifuged at 500 x *g* for 5 min. Cells were resuspended in FACS buffer (0.5% BSA, 2 mM EDTA in PBS) and stained with DAPI 5 min before sorting on a FACSMelody cell sorter (BD Biosciences).

**Murine primary brain ECs** were isolated from male and female mice as previously described[67]. Briefly, after cutting off the cerebellum and olfactory bulbs, the meninges were removed by rolling the brain on cellulose chromatography paper (WHA3030917; Sigma-Aldrich). Brains were then homogenized in 5 ml digestion buffer (0.2% Collagenase II + 0.05% Dispase II (17105-041; Gibco) + DNase I (DN25; Sigma-Aldrich) in DMEM) using a tissue grinder (D9063; Sigma-Aldrich) and the resulting tissue suspension was further digested in a 37 °C water bath for 30 min while shaking. After stopping the reaction by adding 1 ml FBS, cells were centrifuged and re-suspended in 5 ml 25% BSA-containing DMEM. After centrifugation at 2000 x g for 20 min at 4 °C, the myelin layer was discarded. **For direct cell culture**, pellets were resuspended in EBM-2 medium containing the EGM2 MV supplement (Lonza CC-4147: 5% FBS, 0.04% hydrocortisone, 0.4% hFGF-B,

0.1% VEGF, 0.1% R3-IGF-1, 0.1% ascorbic Acid, 0.1% hEGF, and 0.1% GA-1000) (=EGM2 MV medium). Cells from individual brains were seeded into wells of a 6-well plate and puromycin (P8833, Sigma-Aldrich) was added to reach a final concentration of 8 µg/ml for selection of EC. On days 1 and 2 after seeding, cells were washed twice with PBS, and fresh puromycin-containing culture medium was added. From day 3 on, EGM2 MV medium was replaced every 3 days until cells reached confluency around day 10. **For sorting of brain ECs by FACS**, cell pellets were resuspended in 2 ml red blood cell lysis buffer (130-094-183; Miltenyi) and incubate at RT for 2 min. Following centrifugation at 500 x *g* for 5 min, cells were washed with PBS and resuspended in FACS buffer and stained with antibodies against CD31 (558738; BD Biosciences) and CD45 (12-0451-82; Thermo Scientific) for 30 min on ice. Cells were washed with FACS buffer and stained with DAPI 5 min before sorting. The CD31-positive, CD45-negative population was sorted directly into mRNA lysis buffer using a FACSMelody cell sorter (BD Biosciences).

**Human polymorphonuclear granulocytes**, Human polymorphonuclear granulocytes (PMN) were isolated from healthy donors (male and female; age 25–40 years) and experiments were approved by the local ethics committee of the Hessian Regional Medical Board (Ethikkommission des Fachbereiches Medizin der Goethe-Universität Frankfurt; AZ 110/11) and informed consent was obtained from all donors. Briefly, 5 ml human peripheral blood was collected and layered over separation media (Lympholyte Poly; CL5071; Cedarlane labs). After centrifugation at room temperature for 35 min with 500 x *g*, the PMN containing layer was collected. 2 ml 1 x red blood cell lysis buffer (420301; BioLegend) were used to resuspend the cell pellet. After washing twice in HBSS (14175129; Thermo Scientific), PMN were kept in HBSS containing 2% human serum albumin (H4522; Sigma-Aldrich) for further studies.

## Cell transfection

For **knockdown experiments**, cells were transfected with siRNAs directed against human GPR153 (SASI_Hs01_00110231, Sigma-Aldrich, NM_207370, target sequence: 5′- CUGUGCUGGUGGUGAGCUU-3′ or ESIOPEN, Sigma-Aldrich, ENSG00000158292), human GNAS (SASI_Hs01_00246751, Sigma-Aldrich, NM_000516, target sequence: 5′-CUGAUUGACUGUGCCCAGU-3′)or respective negative control siRNAs (MISSION® siRNA Universal Negative Control #1, SIC001, Sigma-Aldrich or Mission esiRNA, EHUEGFP, Sigma-Aldrich) using Opti-MEM (31985062, Thermo Scientific) and Lipofectamine RNAiMAX (13778150, Invitrogen) for 6 h on two consecutive days. If not otherwise indicated, cells were used 48 h after 2nd transfection.

**Transient transfection of HEK293T cells with an GPR153 expression vectors** (RC206707, Origene, NM_207370) were carried out with Opti-MEM (31985062, Thermo Scientific) and Lipofectamine 2000 transfection reagent (11668019, Thermo Scientific) following the manufacturer's protocol.

## Apoptosis assay

hAoSMCs were seeded onto 96-well plates with $1 \times 10^4$ cells per well and transfected with siRNA for 48 h. After another 48 h of serum starvation, cells were trypsinized and Annexin V staining was performed according to the manufacturer's protocol (R37176, Thermo Scientific). To stain dead cells, DAPI (1 µg/mL, D3571, Thermo Scientific) or 7-AAD (0.5 µg/ml, 559925, BD Biosciences) was added to samples 5 min before analysis on a FACS Canto II flow cytometer.

## Adhesion assay

$0.8 \times 10^4$ HUVECs were seeded per well of a 96-well plate and transfected with siRNA for 48 h. Cells were then treated with 10 ng/ml TNFα or 1 µg/ml LPS treatment for the indicated times. THP-1 cells were labeled with 0.5 µM CellTrace CFSE (C34554, Thermo Fisher) for 30 min at 37 °C, washed with PBS, and resuspended at a concentration

of $10^6$ cells in EC medium. Labeled THP-1 cells were then co-cultured with HUVECs for 30 min at 37 °C, then non-adherent cells were removed by washing with EC medium 3 times. Cells were trypsinized, and the number of CFSE-positive cells was determined using a FACS Canto II flow cytometer.

## cAMP determination

**For GloSensor-based determination of cAMP levels**, HEK cells were transfected with siRNA against human GPR153 or control siRNA (Sigma-Aldrich, SIC001). After 24 h, cells were transfected with the GloSensor™ 22 F plasmid (Promega, E2301) and plasmids encoding GPR153 or empty vector (0.1 µg each plasmid). After 16–20 h, cells were trypsinized and seeded onto 96-well plates with $4 \times 10^4$ cell per well. After another 16–20 h, the medium was replaced with serum-free DMEM containing 10 µM 3-Isobutyl-1-methylxanthine (IBMX, 2845, Bio-Techne GmbH) and 3 mM luciferin D (122799, PerkinElmer). After 2 h, 1 µM of FSK or PGE$_2$ was added and luminescence was measured for 30 min by Flexstation 3 (Molecular Devices). The area under the curve of the luminescence traces was calculated by using SoftMaxPro software (Molecular Devices).

**For the examination of cAMP levels in HEK cells by ELISA**, cells were transfected with siRNA against human GPR153, human GNAS, or control siRNA. After 24 h, cells were transfected with plasmids containing cDNA encoding GPR153 or empty vector (0.1 µg each). The next day, cells were trypsinized and seeded onto 96-well plates with $4 \times 10^4$ cell per well. 48 h after transfection, cells were starved with serum-free DMEM for 2 h, followed by 30 min of 50 µM of IBMX and then 30 min of 1 µM of FSK (F6886, Sigma Aldrich) or PGE$_2$ (2296, Bio-Techne GmbH). The cAMP levels in cell lysates were measured by using a direct cAMP kit (ADI-900-066, Enzo Lifescience) according to the manufacturer's protocol. Protein concentration of the same samples was determined by BCA protein quantification assay (23227, Thermo Scientific). Data were normalized to protein amounts ((cAMP pmol/ml)/mg of protein). To access the effect of pertussis toxin on the cAMP level, HEK cells were transfected with plasmids containing cDNA encoding GPR153 (0.1 µg) or empty vector. After 16–20 h, 100 µl of cells ($4 \times 10^5$ cell/ml) were seeded onto a 96-well plate. The next day, cells were incubated with serum-free DMEM containing 100 ng/ml of pertussis toxin for 3 h or 16 h, followed by 30 min of 50 µM of IBMX and then 30 min of 1µM of PGE$_2$.

**To determine cAMP levels in HUVECs and hCASMCs**, both cell lines were transfected with siRNA against human GPR153 as described above. 24 h after the last transfection, cells were trypsinized and re-seeded in the respective growth media at a density of $10^4$ cells in 100 µl per well on a 96-well plate (HUVECs) or $5 \times 10^4$ cells in 500 µl per well on a 24-well plate (hCASMCs). After 24 h adhesion, the medium was replaced with starvation medium containing 50 µM IBMX for 30 min. Thereafter, ligands were added for 30 min. cAMP levels in cell lysates were measured and calculated as described above.

## EdU staining in vitro and ex vivo

**hAoSMCs** were seeded onto 96-well plates with $1 \times 10^4$ cells per well and transfected with siRNA for 48 h. Following incubation with 10 µM 5-ethynyl-2′-deoxyuridine (EdU) or DMSO in growth medium for 48 h, cells were trypsinized and DNA-incorporated EdU was detected using the Click-iT™ EdU Alexa Fluor™ 647 flow cytometry assay kit (C10424, Thermo Scientific) and a FACS Canto II flow cytometer. The percentage of proliferating cells was calculated by dividing EdU-positive events by all events.

To detect EdU incorporation in **primary murine SMCs**, $2 \times 10^4$ cells / 250 µl were seeded on 0.2% gelatin-coated microscope cover glasses (0111500, Paul Marienfeld) and cultured for 12 h in 48-well plates in growth medium to allow cell attachment. After this, cells were serum-starved for 48 h, then incubated with growth medium containing 10 µM EdU for 24 h. EdU incorporation was detected using the

EdU Cell Proliferation Kit for Imaging (C10340; Thermo Scientific). For analyses in **HUVECs**, $2 \times 10^4$ cells were seeded on collagen I-coated microscope cover glasses and maintained in 48-well cell culture plates for 12 h in EGM2 medium. Cell were then incubated with growth medium containing 10 μM EdU for 2 h and the EdU signal was detected by imaging using the same kit as mentioned in the primary SMC experiment.

For EdU staining in carotid arteries, harvested arteries were embedded in Tissue-Tek O.C.T. compound (#4583, Sakura), snap-frozen on dry ice, and stored at −80 °C for 1 h before cryosectioning. Tissue blocks were cryosectioned at a thickness of 10 μm, and sections were air-dried. EdU staining was performed using the EdU Cell Proliferation Kit for Imaging following the manufacturer's protocol, including staining for αSMA (1A4, C6198; Sigma) and DAPI. Stained slides were imaged using confocal microscopy, and the number of EdU-positive cells was determined within the αSMA-positive area using ImageJ (NIH).

### Expression analysis

**RNA isolation and reverse transcription** were performed using the Quick RNA Micro kit (R1051; Zymo) and ProtoScriptII (E6560; New England Biolabs) according to the manufacturer's instruction. Gene expression levels were determined by real-time PCR using SYBR green PCR mix (Roche) using a LightCycler 480 Probe Master System (Roche). Gene expression was calculated using the ΔΔCt method and normalized to reference gene *GAPDH/Gapdh* (Table 1).

**For RNA sequencing**, EGFP-expressing SMCs were sorted directly into mRNA lysis buffer as described in section 2Isolation and culture of primary cells", and RNA was extracted with RNeasy micro kit (74004; Qiagen). RNA and library preparation integrity were verified with the LabChip Gx Touch 24 (Perkin Elmer). 1 μg of total RNA was used as input for VAHTS Stranded mRNA-seq V6 Library preparation following manufacture's protocol (Vazyme). Sequencing was performed on NextSeq2000 instrument (Illumina) with 1x72bp single-end setup. Trimmomatic version 0.39 was employed to trim reads after a quality drop below a mean of Q15 in a window of 5 nucleotides and keeping only filtered reads longer than 15 nucleotides[68]. Reads were aligned versus Ensembl human genome version hg38 (Ensembl release 109) with STAR 2.7.11a[69]. Alignments were filtered to remove: duplicates with Picard 3.1.1 (Picard: A set of tools (in Java) for working with next generation sequencing data in the BAM format), multi-mapping, ribosomal, or mitochondrial reads. Gene counts were established with featureCounts 2.0.4 by aggregating reads overlapping exons on the correct strand, excluding those overlapping multiple genes[70]. The raw count matrix was normalized with DESeq2 version 1.36.0 (Love et al., Moderated estimation of fold change and dispersion for RNA-Seq data with DESeq2).

**Analysis of previously published data sets:** The expression frequency of orphan GPCRs in SMCs was analyzed in previously published murine[23,26,27] and human[28,29] data sets. SMCs were either defined as *Myh11*-positive, *Cdh5*-negative cells[23,27–29], or, when *Myh11*-expression was too low, by author-provided annotation "SMC"[26]. For analyses in human samples[28,29], cellxgene (https://github.com/chanzuckerberg/cellxgene) and cellxgene VIP were used.

### FACS analysis of spinal cord-infiltrating cells

For the analysis of spinal cord-infiltrating leukocytes by flow cytometry, spinal cord samples were collected 14 days after EAE induction (onset of symptoms). Samples were minced and digested in 1 mL digestion buffer containing 1 mg/mL collagenase D and 50 μg/mL DNase I in RPMI medium at 37 °C for 1 h. The digested tissue was passed through a 100 μm cell strainer to obtain single-cell suspensions. Cells were washed and resuspended in 4 mL cold 90% Percoll, followed by the addition of 3 mL 60% Percoll, 4 mL 40% Percoll, and 3 mL HBSS to create a density gradient. Following centrifugation at 800 × g for

20 min at 4 °C, the upper 4 mL of the gradient were discarded. The remaining solution was collected, leaving 4 mL of Percoll gradient. The collected cells were washed with HBSS, centrifuged at 500 × g for 10 min at 4 °C, and resuspended for staining with the following antibodies: CD45 PE (#12-0451-82), CD19 eFluor 450 (#48-0199-42), F4/80 APC-eFluor 780 (#47-4801-82) and CD4 PerCP-Cyanine5.5 (#45-0042-80) from Thermofisher; Ly6G FITC (#561105) from BD Bioscience; and CD11b Brilliant Violet 510 (#101245), Ly6C PE/Cyanine7 (#128017) and CD8a APC (#100711) from Biolegend. Cells were analyzed using a FACS Canto II (BD Biosciences); gating strategies are shown in the supplement.

### Histological analyses

For the analysis of neointima formation, carotid arteries were fixed with 4% paraformaldehyde in PBS, then embedded in paraffin. Serial sections (5 μm) were collected transversely through the artery using a rotary microtome (Microm HM355S; Thermo Scientific), air-dried, and mounted on SuperFrost Plus slides. After deparaffinization and rehydration, the collected sections were stained using an elastic stain kit (HT25A-1KT; Sigma-Aldrich). Images were acquired with an M165 FC microscope (Leica) and analyzed with Fiji ImageJ (NIH).

For the analysis of demyelination in EAE, vertebral columns were harvested at day 28 after EAE induction and fixed in 4% paraformaldehyde (PFA) overnight, then spinal cords were dissected, dehydrated, cut into pieces of approximately 1 cm length, and embedded in paraffin. Lumbar spinal cords were sectioned using a sliding microtome at a thickness of 10 μm. For luxol fast blue staining, dewaxed slides were washed in 95% ethanol and stained with 0.1% LFB solution (prepared by dissolving 0.1 g LFB (10348250; ThermoFisher) in 100 mL of 95% ethanol and 0.5 mL acetic acid) for 10 h at 56 °C. After rinsing in 95% ethanol, slides were differentiated in 0.05% lithium carbonate solution for 30 s, followed by several washes in distilled water. Slides were then ethanol-dehydrated, coverslipped with PERTEX (VWRKAM-0801; Avantor), and sectiond centering around the L5 vertebral segment were analyzed under a bright light microscope. The pictures were quantified with Fiji ImageJ.

Quantification of infarct volumes after MCAO was performed in coronal brain sections using a silver staining technique as described previously[71].

### Immunofluorescence staining in vitro and ex vivo

**To determine MRTF-A localization**, hCASMCs ($2 \times 10^4$/250 μl) were seeded onto poly-L-lysine-coated microscope cover glasses in 48-well cell culture plates and allowed to attach for 12 h. Cells were then washed with PBS and fixed for 15 min with 250 μl of 3.7% formaldehyde. Permeabilization was performed in each well with 250 μl 0.5% Triton X-100 in PBS for 20 min. Then, cells were blocked with blocking buffer (3% BSA, 0.5% Triton X-100, PBS) at RT for 30 min. Primary antibodies dissolved in blocking buffer (1:500, 200 μl per well) were incubated overnight with shaking at 4 °C. After 3 times of washing with PBS, secondary antibodies, dissolved in blocking buffer (1:1000, 200 μl per well), were incubated with the cells at RT for 1 h, protected from light. The cover glasses with stained cell samples were then collected and mounted on SuperFrost Plus slides (J1800AMNZ, Epredia) with Fluoromount solution (21634.01, SERVA). Images were captured and processed by confocal microscopy (Leica TCS SP5). Quantification was performed by taking 3–6 pictures from each slide followed by analysis with ImageJ (NIH), and individual cells were classified as predominantly nuclear MRTF-A, predominantly cytosolic MRTF-A or no clear preference.

**To detect CREB phosphorylation in carotid arteries**, vessels were collected from mice at day 7 after ligation. Tissues were fixed in 4% PFA overnight, cryosectioned at 10 μm thickness, and air-dried. After washing with PBS, sections were permeabilized and blocked in blocking buffer (3% BSA, 0.5% Triton X-100 in PBS) as described above.

**Table 1 | Primer sequences qRT-PCR**

| Gene | Forward Primer | Reverse Primer |
|---|---|---|
| **Human primers** | | |
| ACTA2 | ACCCGATAGAACATGGCATCA | GGCAACACGAAGCTCATTGTA |
| ALPL | ACTGGTACTCAGACAACGAGAT | ACGTCAATGTCCCTGATGTTATG |
| ANKRD1 | GATCGAATTCCGTGATATGCT | AAACATCCAGGTTTCCTCCA |
| BCL2 | GGTGGGGTCATGTGTGTGG | CGGTTCAGGTACTCAGTCATCC |
| BDNF | GGCTTGACATCATTGGCTGAC | CATTGGGCCGAACTTTCTGGT |
| BGN | GAGACCCTGAATGAACTCCACC | CTCCCGTTCTCGATCATCCTG |
| BIRC5 | AGGACCACCGCATCTCTACAT | AAGTCTGGCTCGTTCTCAGTG |
| CCL2 | CAGCCAGATGCAATCAATGCC | TGGAATCCTGAACCCACTTCT |
| CCL3 | GCAACCAGTTCTCTGCATCA | AAGATGACACCGGGCTTG |
| CCL5 | CCAGCAGTCGTCTTTGTCAC | CTCTGGGTTGGCACACACTT |
| CCN2 | GCCTCCTGCAGGCTAGAGA | GATGCACTTTTTGCCCTTCT |
| CCNA2 | CCATACCTCAAGTATTTGCCATC | TCCAGTCTTTCGTATTAATGATT |
| CCND1 | CAATGACCCCGCACGATTTC | CATGGAGGGCGGATTGGAA |
| CD68 | GGAAATGCCACGGTTCATCCA | TGGGGTTCAGTACAGAGATGC |
| CREM | ACACCACCTAGTATTGCTACCA | GGATTGTTCCACCTTGGGCTAT |
| CYR61 | TCCAGGGCACACCTAGACA | GCCCATTTTCTCCATGATTC |
| FGF2 | AGTGTGTGCTAACCGTTACCT | ACTGCCCAGTTCGTTTCAGTG |
| GAPDH | GAACGGGAAGCTTGTCATCAA | ATCGCCCCACTTGATTTTGG |
| GNAS | TGCCTCGGGAACAGTAAGAC | GCCGCCCTCTCCATTAAAC |
| GPR153 | CGTGTCCACCTTCTACACCC | GCTCAGCCGGTAGTTGACAG |
| ICAM1 | CCCCTACCAGCTCCAGACC | TGCGTGTCCACCTCTAGGAC |
| IL1b | ATGATGGCTTATTACAGTGGCAA | GTCGGAGATTCGTAGCTGGA |
| IL6 | AGAGCTGTGCAGATGAGTACAA | GTTGGGTCAGGGGTGGTTA |
| ITGAM | ACTTGCAGTGAGAACACGTATG | TCATCCGCCGAAAGTCATGTG |
| LUM | GGATTGGTAAACCTGACCTTCAT | GATAAACGCAGATACTGCAATGC |
| MKI67 | CAGAGAATTTGCTTGGAAAACA | GAGGTGGGGAGCAGAGGT |
| MYC | GGCTCCTGGCAAAAGGTCA | CTGCGTAGTTGTGCTGATGT |
| MYH11 | CTGCAGCAGCTCTTCAACC | CCTCGCGCTGGTACTCCT |
| NR4A1 | ATGCCCTGTATCCAAGCCC | GTGTAGCCGTCCATGAAGGT |
| PCNA | CCTAAAGATTCTGAAAAAGAGAA | CAGGTTGCAAAGGACATGC |
| PDGFB | CTCGATCCGCTCCTTTGATGA | CGTTGGTGCGGTCTATGAG |
| PTGS2 | CTGGCGCTCAGCCATACAG | CGCACTTATACTGGTCAAATCCC |
| RUNX2 | TCAACGATCTGAGATTTGTGGG | GGGGAGGATTTGTGAAGACGG |
| SPP1 | GAAGTTTCGCAGACCTGACAT | GTATGCACCATTCAACTCCTCG |
| TAGLN | GGCCAAGGCTCTACTGTCTG | GCCATAGGAAGGACCCTTGT |
| TNF | CCCTGGTATGAGCCCATCT | CTGAGTCGGTCACCCTTCTC |
| VCAM1 | CAGGCTGGAAGAAGCAGAAA | CACTCTCAGAAGGAAAAGCTGTA |
| VEGFA | AGGGCAGAATCATCACGAAGT | AGGGTCTCGATTGGATGGCA |
| **Murine primers** | | |
| Acta2 | GAGGCACCACTGAACCCTAA | TACATGGCGGGGACATTGAA |
| Actb | CCCTAAGGCCAACCGTGAAA | CAGCCTGGATGGCTACGTAC |
| Ccl2 | CATCCACGTGTTGGCTTA | GATCATCTTGCTGGTGAATGAGT |
| Ccl5 | GCTGCTTTGCCTACCTCTCC | TCGAGTGACAAACACGACTGC |
| Cdh5 | AACGAGGACAGCAACTTCAC | TGGCATGCTCCCGATTAAAC |
| Fgf2 | CGGCTCTACTGCAAGAACG | TGCTTGGAGTTGTAGTTTGACG |
| Gapdh | AGCTTGTCATCAACGGGAAG | TTTGATGTTAGTGGGGTCTCG |
| Gpr153 | CGATGGCCAGTCAACTACC | CCGACAGGATGAAGGACAC |
| Icam1 | AGGGCTGGCATTGTTCTCTA | TGTCGAGCTTTGGGATGGTA |
| Il-6 | CGATGATGCACTTGCAGAAA | ACTCCAGAAGACCAGAGGAA |
| Mki67 | GAGACATACCTGAGCCCATCA | GCTTTGCTGCATTCCGAGTA |
| Myh11 | AAGCTGCGGCTAGAGGTCA | CCCTCCCTTTGATGGCTGAG |
| Pcna | TTTGAGGCACGCCTGATCC | GGAGACGTGAGACGAGTCCAT |
| Tagln | ACCAAAAACGATGGAAACTACCG | GTGAAGTCCCTCTTATGCTCCT |
| Tgfb1 | TGGAGCAACATGTGGAACTC | CAGCAGCCGGTTACCAAG |

**Table 1 (continued) | Primer sequences qRT-PCR**

| Gene | Forward Primer | Reverse Primer |
|------|----------------|----------------|
| *Tnf* | GGGTGATCGGTCCCCAAA | TGAGGGTCTGGGCCATAGAA |
| *Vcam1* | CCCAAACAGAGGCAGAGTGTA | CTGCAGGATGCATTGGTACAC |

Incubation with rabbit anti-phospho CREB (Ser133) (#9198) antibodies was performed at 1:500 dilution in blocking buffer overnight at 4 °C. After three times PBS washes, sections were incubated with anti-rabbit IgG Alexa Fluor 488 (#A21206; Thermofisher), anti-αSMA-Cy3 (C6198; Sigma), and DAPI, for 1 h at RT in the dark. Sections were washed with PBS, mounted with Fluoromount solution, and analyzed using confocal microscopy. Phospho CREB signals within the αSMA-positive area were quantified using ImageJ.

**Stainings on MCAO tissues** were performed on cryosections after 10 min of methanol fixation (-20 °C). After 30 min of blocking (1% BSA, PBS), sections were incubated with respective primary antibodies (in blocking solution) overnight at 4 °C. After washing, secondary antibodies were incubated at room temperature for 1.5 h before mounting. Four images per region per mouse were taken with a Leica fluorescence microscope (AF6000). Analysis was performed blinded to the genotype. The following antibodies were used: anti-VCAM1 (#550547) and anti-Ly-6G (#551459) were purchased from BD-Pharmingen, antibodies against anti-collagen IV (#134001) were from Bio-Rad.

### Permeability assay

HUVECs were seeded onto 96-transwell plates with $0.8 \times 10^4$ cells per well and transfected with siRNA for 48 h. Before the experiment, the monolayer was visually checked by CFSE staining, and only wells with 100% confluency were used. After treatment with 10 units/ml thrombin for 10 min, the drug-containing medium in the transwell insert was replaced with 50 µl of EGM2 medium containing 1 mg/ml FITC-Dextran (70 kDa, D1822, Thermo Fisher), and the lower chambers were filled with 250 µl EGM2 medium. At the indicated time points, 5 µl of solution from the lower chamber was transferred to a 96-well reading plate containing 100 µl PBS per well (for time point 0, 5 µL of EGM2 medium were used), and the fluorescence signal was measured at 494/518 nm using Flexstation 3 (Molecular Devices).

### Proliferation assays

48 h after siRNA transfection, hCASMCs or hAoSMCs were seeded onto 96-well plates with $1 \times 10^4$ cells per well and allowed to attach for 12 h in growth medium. After this, cells were further cultured in growth medium or growth medium + 10 ng/ml PDGF-BB (medium exchange every 24 h), and cell numbers were analyzed at the indicated time points after seeding. To do so, cells were stained with 5 µM CFSE (for 30 min), then individual wells were trypsinized and total cell numbers were detected as CFSE-positive events using a FACS Canto II flow cytometer (BD Biosciences).

### Transwell migration assays

HUVECs were seeded onto 6-well plates with $1.6 \times 10^5$ cells per well and transfected with siRNA for 48 h. Cells were then trypsinized and $1.2 \times 10^4$ cells were seeded into the collagen I-coated upper compartment of a 96-transwell plate (3388; Corning). After overnight adhesion, the integrity of the endothelial monolayer was visually checked by CFSE staining, and only wells with 100% confluency were used. Where indicated, HUVECs were preincubated with TNFα (10 ng/ml) or LPS (1 ug/ml) for 4 h. Cells in the upper compartment were washed and co-cultured with CFSE-labeled THP-1 cells or primary human PMN ($8 \times 10^5$/ml) in EGM2 medium. In all experiments, 250 µl culture medium with or without chemoattractant was added to the lower compartment, and CFSE-labeled cells were allowed to transmigrate for 6 h at 37 °C and 5% $CO_2$. After that, transwell plates were kept on ice for 20 min, then inserts were discarded and cells in the lower compartment collected. The number of THP1 cells transmigrated compared to the input was determined by flow cytometry.

### Western blotting

The following antibodies were used: anti-phospho-LATS1/2 (Ser909/Ser872) (#PA5-64591) and anti-KLF4 (#11880-1-AP) from Thermofisher, anti-MRTF-A (H-140) (#sc-32909) from Santa Cruz, anti-VCAM1 (EPR5047, #ab134047), TAGLN (#ab14106) and anti-α-smooth muscle actin (#ab5694) from Abcam. Antibodies directed against phospho-MOB1 (Thr35) (D2F10, #8699), YAP/TAZ (D24E4, #8418), phospho-YAP (Ser127) (#4911), CREB (48H2, #9197), phospho-CREB (Ser133) (87G3, #9198), AKT (#9272), phospho-AKT (Ser473) (D9E, #4060), Erk1/2 (137F5, #4695), phospho-Erk1/2 (Thr202/Tyr204) (D13.14.4E, #4370), NF-κB (C22B4, #4764), phospho-NF-κB (Ser536) (93H1, #3033), phospho-SMAD2 (Ser465/467)/SMAD3 (Ser423/425) (D27F4, #8828), SMAD2/3 (D7G7, #8685), phospho-Stat1 (Tyr701) (58D6, #9167), Stat1 (#9172), phospho-Stat3 (Ser727) (#9134), Stat3 (124H6, #9139), Lamin A/C (#2032), α-Tubulin (11H10, #2125) and GAPDH (14C10, #2118) were obtained from Cell Signaling.

**For analyses in HEK cells**, cells were seeded into 6-well plates ($0.3 \times 10^6$ cells/well) and transfected with siRNA against human GPR153 or a negative control. After 24 h, cells were transfected with plasmids encoding GPR153 (0.1 µg) or empty vector. After 48 h, the medium was replaced with serum-free DMEM for 2 h, followed by addition of 1 µM of FSK or $PGE_2$ for 10 min. Cells were washed with cold-PBS and lysed with RIPA buffer (100 mM Tris-HCl pH 7.5, 5 mM EDTA, 50 mM NaCl, 50 mM β-glycerophosphate, 50 mM NaF, 0.1 mM $Na_3VO_4$, 0.5% NP-40, 1% TritonX-100, 0.5% sodium deoxycholate) supplemented with protease/phosphatase inhibitors (Thermo Fisher Scientific, 78445). Proteins were separated by sodium dodecyl sulfate-polyacrylamide gel electrophoresis and transferred onto nitrocellulose membranes (Cytiva, 10600003). After blocking in 5% (w/v) BSA for 1 h at room temperature, membranes were incubated overnight at 4 °C with primary antibodies. After washing 3 times with TBS-T for 15 min, membranes were incubated with HRP-conjugated anti-rabbit IgG (1:1000, Cell Signaling Technology, 7074) for 1 h at room temperature. Immunoblots were visualized by chemiluminescence reagent (Millipore, WBKLS0500) and ChemiDoc MP Imaging System using Image Lab™ Software (Bio-Rad). Densitometric analyses were done using Fiji software (NIH). After background subtraction, data for individual antibodies were normalized to GAPDH as internal reference gene.

**For analyses in human or murine ECs and SMCs**, confluent cells in a 24-well plate (around $2 \times 105$ cells per condition) were lysed with RIPA buffer (Pierce RIPA Buffer, 89900, Thermo Scientific) supplemented with protease inhibitor cocktail (1861281, Thermo Scientific). After determining protein concentration by BCA assay (23227, Thermo Scientific), samples were boiled at 95 °C for 10 min with Laemmli and subjected to sodium dodecyl sulfate-polyacrylamide gel electrophoresis. Protein was then transferred to nitrocellulose membranes (GE10600002; Sigma-Aldrich), followed by overnight incubation with primary antibodies in 4 °C. After washing 3 times with TBS-T for 5 min, membranes were incubated with horseradish peroxidase-conjugated (HPC) conjugated secondary antibodies directed against rabbit or mouse IgG (1:3000, #7074 or #7076; Cell Signaling Technology) for 2 h

at room temperature and were developed using the ECL detection system (32106; Thermo Scientific). Protein band intensities were analyzed by ImageJ software.

## Wire myography

Tamoxifen-treated mice were sacrificed with $CO_2$, and first-order mesenteric arteries were dissected. Artery segments of 2 mm length were mounted in wire myograph chambers (Danish Myo Technology, 610 M) and kept in freshly prepared Krebs-Henseleit buffer with carbogen gas aeration at 37 °C. After a 30 min recovery period and normalization, we determined contractile responses by cumulative administration of the indicated agonists. Responses to relaxants cicaprost (Cayman, 16831), sodium nitroprusside (Fluka, 71780), and acetylcholine (Sigma, A6625) were measured after precontraction with phenylephrine (10 μM, Sigma, P6126) and are expressed as percentages of maximal contraction; responses to contractile stimuli phenylephrine (10 μM, Sigma, P6126) and U46619 (Cayman, 16450) were normalized to reference contractions elicited by 60 mM $K^+$ Krebs solution.

## Viability assay

To determine cell viability, live-cell protease activity was measured using the CellTiter-Fluor™ cell viability assay kit (G6080, Promega). To do so, cells from different groups were seeded into 96-well plates at a density of $1 \times 10^4$ cells per well in growth medium and allowed to attach for 12 h. After this, cell were serum-starved for 48 h, and the viability assay reagent was added in an equal volume during the last 30 min incubation. The resulting fluorescence was measured using a fluorometer (FlexStation 3 Microplate Reader, Molecular Devices).

## General statistical analyses

If not otherwise indicated, data are presented as means ± standard errors of the means (SEM). Data were tested for normal distribution using the Shapiro-Wilk test or the Kolmogorov−Smirnov test. The following statistical tests were used for normally distributed samples: unpaired or paired student's $t$-test for comparisons between two groups, one-sample t-test for comparisons between test group and a designated value, one-way ANOVA with Dunnett's post hoc test for comparison of multiple groups, two-way ANOVA with Sidak's posthoc test for comparisons between two groups at different times or treatments. For data sets that did not follow normal distribution, Mann-Whitney test was used for the comparison of two groups, in some cases followed by the two-stage step-up method (Benjamini, Krieger, and Yekutieli) to adjust for multiple testing. n refers to the number of independent experiments or mice per group. The number of mice per group was determined based on power calculations for the primary parameter with mean differences and standard deviations taken from previous studies, and a power of 80% with a standard level of significance of 0.05. Statistical analyses were performed using GrahPad Prism 10 (version 10.1.2).

## Reporting summary

Further information on research design is available in the Nature Portfolio Reporting Summary linked to this article.

# Data availability

The mRNA sequencing data have been deposited in the GEO database under accession code GSE270457. All other data supporting the findings of this study are available within the paper and its Supplementary Information. Source data are provided with this paper.

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

## Acknowledgements

We thank Martina Winkels, Ulrike Krüger, and Claudia Kopp for expert technical assistance. This work was funded by the *Deutsche Forschungsgemeinschaft* (DFG, German Research Foundation) through projects 456687919–SFB 1531 (A04 to NW, A03 to RB, B01 to SD, A06 to SO), DFG WE2891/2-1 (to NW), DFG SCHW 416/12-1 (to MS), and the LOEWE-GLUE initiative LOEWE/2/12/519/03/05.001(0014)/71 (to NW).

## Author contributions

J.S. performed most experiments and wrote parts of the manuscript. He was supported by J.K. (HEK experiments), T.W. (myography), S.G. (RNA sequencing), Z.S., and J.W. (MCAO). L.S.T., T.W., R.P.B., and S.D. provided single-cell sequencing data. S.O. and M.S. provided mice, interpreted data, and reviewed the manuscript, N.W. designed and supervised the study, analyzed data, and wrote the manuscript.

## Funding

## Competing interests

The authors declare no competing interests.
