## [Peer Review File · Nature Communications]

Orphan receptor GPR153 facilitates vascular damage responses by modulating cAMP levels, YAP/TAZ signaling, and NF- κ B activation

Corresponding Author: Professor Nina Wettschureck

Version 0:

Reviewer comments:

Reviewer #1

(Remarks to the Author)

In this work, the authors mainly found that GPR153 was upregulated in smooth muscle cells (SMCs) in response to injury; Knockdown of GPR153 resulted in reduced proliferation and mildly altered differentiation in human SMCs; SMC-specific GPR153 deficiency mice could partially protect against ligation-induced neointima formation; GPR153 deficiency hCASMCs resulted in increased cellular cAMP levels and increased CREB phosphorylation, reduced YAP/TAZ levels, and inhibited NF- κ B activation; EC-specific GPR153 deficiency mice also resulted in reduced inflammatory gene expression. The reviewer has the following comments.

Major:

1. Just as the authors suggested that "Numerous pathways have been implicated in these processes, but we focused on players known to be regulated by GPCRs, such as ERK1/2, AKT, YAP/TAZ, NF- κ B, CREB or MRTF. GPR153 knockdown did not alter the phosphorylation state of ERK1/2, AKT, or NF- κ B p65 (Suppl. Fig. 2A-D), but phosphorylation of YAP was enhanced (Fig. 5A,B)". "In addition to altered YAP/TAZ levels, we found CREB phosphorylation to be significantly elevated (Fig. 5F-I), and also expression of select CREB target genes was increased (Fig. 5J)." Except these "players known to be regulated by GPCRs", other proteins or signaling pathways might be more important to deserve studying.
2. Again, as the authors conclude that " Loss of GPR153 led to both in SMCs and ECs to disinhibition of cAMP-dependent signaling, resulting in reduced NF- κ B and YAP/TAZ signaling as well increased CREB-dependent gene expression. In EC, this results mainly in reduced inflammatory activation and consecutively reduced immune cell recruitment in models of neuroinflammation, whereas in SMC, reduced proliferation and dedifferentiation protected mice from adverse vascular remodeling in models of neointima formation". Is there any interaction between the SMCs and ECs?
3. If orphan receptor GPR153 facilitates pro-inflammatory and pro-proliferative gene expression in ECs and SMCs by controlling cellular cAMP levels. cAMP, the second messenger, could be controlled by many GPCRs. It is difficult for me to understand there is differences between GPR153 and other GPCRs controlled cAMP.

Reviewer #2

(Remarks to the Author)

Please see the attached.

Reviewer #3

(Remarks to the Author)

In this data-enriched submission, the authors described their investigation of the functions of the orphan receptor GPR153 in vascular cells. They generated smooth muscle cell-specific or endothelial cell-specific GPR153 conditional knockout mouse lines, which is a strength of the study. They conclude that GPR153 facilitates pro-inflammatory and pro-proliferative vascular damage responses by regulating vascular smooth muscle cells and endothelial cells, particularly the cAMP/CREB signaling. Unfortunately, the manuscript is a collection of descriptive datasets generated using various cell cultures and

animal models without focused scientific hypotheses being tested. For instance, the last figure describes three disease models (EAE, stroke, and TBI); however, the data are too preliminary to draw meaningful conclusions. The manuscript is extremely difficult to follow, as the logical flow is unclear to the readers.

Specific critiques:

1. The first half of the study (Figs. 1-5) investigated the role of GPR153 in vascular smooth muscles using cell cultures and an in vivo carotid artery occlusion model. The main finding in cell cultures was that GPR153 knockdown inhibits the proliferation of smooth muscle cells under basic conditions or in the presence of growth factors. The functional significance of such an effect is unclear.

They later studied the signaling pathway under GPR153 regulation using human coronary artery smooth muscle cells. They found that GPR153 knockdown upregulates CREB, YAP/TAZ, and NF- κ B signaling. Again, the functional significance of this pathway is unclear. No follow-up experiments were performed to determine the functional consequence of manipulating this signaling pathway.

2. They generated SMC-specific GPR153 cKO mice and performed the carotid artery ligation model. They found that GPR153 cKO inhibits SMC proliferation, differentiation, and neointima formation. However, the data are too preliminary to conclude, as the functional impact of GPR153 is unclear, and the signaling mechanism is unknown.

Using Myh11-CreERT2 mice to generate SMC-specific GPR153cKO mice has several caveats that may affect data interpretation. First, the Myh11-CreERT2 transgene is located on the Y chromosome, which means it can only be used to study male mice. However, sex dimorphism of vascular structure and function has been increasingly recognized. Second, the Myh11-CreERT2 transgene can recombine in vascular and non-vascular smooth muscles. Thus, the effects of GPR153 cKO on non-vascular smooth muscles should be considered.

3. Figure 6 shows the effects of GPR153 knockdown on cellular cAMP levels and pCREB/TAZ signaling. These experiments were done using HEK cells. The significance of the data is unclear, as HEK cells are not vascular cells.

4. They generated EC-specific GPR153 cKO mice and performed injury models using both GPR153 cKO and WT control mice, including the EAE model and a focal brain stroke model (Fig. 8). However, the data are too preliminary to conclude. In the EAE study, essential pathological data are lacking to demonstrate the histological basis for GPR153 cKO-afforded protection. The stroke outcome assessment is limited to the acute stage (48 hours) and lacks a dedicated assessment of neurological functions. The mechanisms by which endothelial GPR153 activation leads to neuronal injury in EAE or stroke are not investigated in the current study.

Version 1:

Reviewer comments:

Reviewer #1

(Remarks to the Author)

The authors have addressed most of my concerns. No further comments.

Reviewer #2

(Remarks to the Author)

In this revised manuscript, the authors have addressed well all the concerns raised from the original version except one as below.

In Fig.6 to address the main concern No. 5 in the rebuttal letter, while the data on panels B, C and D are well depicted, it is hard to understand the purpose of the data on panel A. First, there must be no statistical significance between the samples obtained from Control and iBEC-G153-KO. The second is how this data is different from the one shown in Fig. 8J in the revised manuscript. The third is I have realized that there is no description how the authors obtained "neurological score" in the manuscript.

Reviewer #4

(Remarks to the Author)

To Authors:

This study may offer some fundamental advances in terms of mechanisms. However, the parts that try to apply these pathways to disease models are somewhat diluted.

The effects on SMC proliferation are clear and exciting. So, would it have been nice to focus on this and see how this pathway affects some type of clinical disease like carotid stenoses or coronary artery narrowing?

The effects on endothelial cells are also exciting if it is true that GPR153 mediates NFkB inflammation, upregulates adhesion molecules and then promotes infiltration of damaging immune cells into the CNS. But if so, would it be nice to just pick stroke and go deeper?

- a. For example, what happens to GPR153 and some of the downstream signals in the stroke model itself?
- b. If GPR153 affects cAMP, what happens to vasoregulation and cerebral blood flow in normal or ischemic conditions?
- c. The cell culture studies refer to permeability. What happens to the BBB after focal cerebral ischemia?
- d. The authors show that VCAM1 is decreased and infiltrated immune cells are lower. Is there some spatial correlation of these two events with the GPR153 signals?
- e. If there are effects on demyelination in the EAE model, should one also see decreased immune cells in the corpus callosum after focal cerebral ischemia along with improved white matter integrity?
- f. Finally, in the stroke field, it is somewhat common to perform behavior tests for at least 1-2 weeks post-stroke. Would this approach be useful here?

**Response to reviewers' comments for Shao et al.,
Orphan receptor GPR153 facilitates vascular damage responses by modulating cAMP levels,
YAP/TAZ signaling, and NF- κ B activation**

Reviewer #1 (Remarks to the Author):

In this work, the authors mainly found that GPR153 was upregulated in smooth muscle cells (SMCs) in response to injury; Knockdown of GPR153 resulted in reduced proliferation and mildly altered differentiation in human SMCs; SMC-specific GPR153 deficiency mice could partially protect against ligation-induced neointima formation; GPR153 deficiency hCAsMCs resulted in increased cellular cAMP levels and increased CREB phosphorylation, reduced YAP/TAZ levels, and inhibited NF- κ B activation; EC-specific GPR153 deficiency mice also resulted in reduced inflammatory gene expression. The reviewer has the following comments.

We thank the reviewer for the helpful comments and have addressed all major points mentioned below.

Major:

1. Just as the authors suggested that “Numerous pathways have been implicated in these processes, but we focused on players known to be regulated by GPCRs, such as ERK1/2, AKT, YAP/TAZ, NF- κ B, CREB or MRTF. GPR153 knockdown did not alter the phosphorylation state of ERK1/2, AKT, or NF- κ B p65 (Suppl. Fig. 2A-D), but phosphorylation of YAP was enhanced (Fig. 5A,B)”. “In addition to altered YAP/TAZ levels, we found CREB phosphorylation to be significantly elevated (Fig. 5F-I), and also expression of select CREB target genes was increased (Fig. 5J).” Except these “players known to be regulated by GPCRs”, other proteins or signaling pathways might be more important to deserve studying.

This is a valid point, and following the reviewer's suggestion we have studied a number of other pathways implicated in SMC (de)differentiation that are not directly regulated by GPCRs, for example SMAD phosphorylation,¹ STAT phosphorylation,² or KLF4 expression.³ As before, we studied activation of these signaling cascades 1) in the serum-starved, differentiated state and 2) when cultured in the presence of growth factor-containing medium. We found that phosphorylation of SMAD2/3 and STAT1, as well as total KLF4 levels, were increased in the presence of growth factor-containing medium, but we did not detect significant differences between cells treated with control siRNA (SiContr) or siRNA directed against GPR153 (siGPR153) (Fig. 1A-F). STAT3 phosphorylation, in contrast, was not increased in the presence of growth factor-containing medium, but we observed increased STAT3 phosphorylation in GPR153 knockdown cells (Fig. 1G,H).

Figure 1: A-H, hCASCs transfected with control siRNA (siContr) or GPR153-specific siRNA (siG153) were grown for 48 h in the presence of serum- / growth factor-containing medium (growth medium, GM) or in growth factor-reduced medium (starvation medium, SM), then analyzed by immunoblotting with the indicated antibodies (GAPDH as loading control, n=6-8). Data are means \pm SEM; differences between genotypes were analyzed using two-way ANOVA and Sidak's post-hoc test; *, $P < 0.05$.

The mechanisms underlying the increased STAT3 phosphorylation are not fully clear and literature does not provide evidence for a direct crosstalk between GPR153 or CREB/YAP/TAZ/NFκB signaling on one side and STAT3 phosphorylation on the other side. A previous study using shRNA-mediated knockdown or overexpression of STATs in SMCs indicated that while STAT1 promotes dedifferentiation, STAT3 promotes a more contractile phenotype,² so increased STAT3 phosphorylation might reflect the slightly more differentiated state of GPR153-deficient SMC.

We have added these data as Suppl Fig. 3A-H to the revised manuscript and describe them on page 6, line 25-27.

2. Again, as the authors conclude that “ Loss of GPR153 led to both in SMCs and ECs to disinhibition of cAMP-dependent signaling, resulting in reduced NF-κB and YAP/TAZ signaling as well increased CREB-dependent gene expression. In EC, this results mainly in reduced inflammatory activation and consecutively reduced immune cell recruitment in models of neuroinflammation, whereas in SMC, reduced proliferation and dedifferentiation protected mice from adverse vascular remodeling in models of neointima formation”. Is there any interaction between the SMCs and ECs?

This is an interesting question, since ECs and SMCs are indeed known to interact directly and indirectly in numerous ways. Direct EC/SMC crosstalk involves physical contact through cell surface proteins such as connexins, Eph/ephrins, and Jagged/Notch3, whereas indirect EC/SMC dialogue is mediated by release of extracellular matrix components, extracellular vesicles, or other soluble mediators.⁴⁻⁶ Especially under conditions of vascular damage, soluble inflammatory mediators are upregulated both in ECs and SMCs and can affect the respective other cell populations. Activated ECs, for example, release pro-inflammatory cytokines like IL-6 and CCL2 (aka MCP1), which can enhance SMC migration, proliferation, and inflammatory activation.⁶⁻⁸ The same is true for SMCs, which can influence EC function by releasing inflammatory mediators such as TNFα or IL-1β, which in turn may activate ECs, thereby amplifying the inflammatory response.^{9,10}

Since reduced inflammatory activation is a hallmark of GPR153-deficient SMCs and ECs, we investigated whether soluble mediators implicated in EC/SMC crosstalk were altered in the absence of GPR153. In TNFα-activated GPR153-deficient brain ECs, we found transcript levels for mediators such as TNFα, IL-6, and CCL2/MCP1 to be significantly reduced (Fig. 2A). In line with this, CCL2 ELISA showed reduced secretion of CCL2 from these cells (Fig. 2B). We also studied production of inflammatory mediators in GPR153-deficient SMCs, in this case using hCASMCs as a model. Also here, cytokine-encoding mRNAs such as *IL6* and *IL1b* were reduced, as well as *CCL2* and *PDGF* (encoding platelet-derived growth factor) (Fig. 2C). The reduced IL-6 production was confirmed on the protein levels (Fig. 2D). Together, these data indicate that GPR153-deficient ECs and SMCs indeed produce lower amounts of pro-inflammatory mediators that have the capacity to modulate the activation state of adjacent cells.

Figure 2: **A**, Gene expression was determined by qRT-PCR in TNF α (10 ng/ml, 2h)-stimulated BECs from control mice and iBEC-G153-KOs (data normalized to *Gapdh*, control set to 1, n=8-10). **B**, CCL2 levels in the supernatant of TNF α (10 ng/ml, 4h)-stimulated BECs from control mice and iBEC-G153-KOs was determined by ELISA (n=4). **C**, Gene expression was determined by qRT-PCR in growth medium-cultured hCASMCs after siRNA-mediated knockdown of GPR153 (data normalized to *GAPDH*, control set to 1, n=6). **D**, IL-6 levels in the supernatant of growth medium-cultured control and GPR153-kd hCASMCs were determined by ELISA (n=8). Data are means \pm SEM; differences between genotypes were analyzed using two-way ANOVA and Sidak's post-hoc test (A,C) or unpaired t test (B,D). *, $P < 0.05$; **, $P < 0.01$; ***, $P < 0.001$; ****, $P < 0.0001$.

We also investigated whether reduced production of inflammatory mediators in ECs or SMCs was able to change the differentiation state of the respective other cell type in a co-culture system. To do so, we used transwell assays to culture ECs and SMCs in two different settings (Fig. 3A,C):

- Setting A: TNF α -treated BECs from control and KO mice cultured in the upper compartment, murine aortic SMCs (maoSMC) from wild-type mice cultured in the lower compartment (Fig. 3A).
- Setting B: Growth-medium stimulated control or GPR153 knockdown hCASMCs in the upper compartment, HUVECs in the lower compartment (Fig. 3C).

After 8 h of co-culture, cells in the lower well were harvested and expression of differentiation/activation markers was analyzed by qRT-PCR. In setting A, both WT and GPR153-KO BECs increased *Vcam1* expression in maoSMC, though the effect was significantly weaker in cells co-cultured with GPR153 knockdown BECs (Fig. 3B). Effects on other marker genes were weak, suggesting that inflammatory mediators secreted from BECs are too diluted in the transwell system to induce wide-spread effects. In setting 2, co-culture with control or GPR153-deficient hCASCs induced some upregulation of inflammatory markers such as *ICAM1*, *Il6*, or *IL1b* in HUVECs, but no clear difference between control and GPR153-deficient cells in the upper chamber were observed (Fig. 3D).

Figure 3: A,B, Gene expression was determined by qRT-PCR in murine aortic SMCs (maoSMCs) that were co-cultured for 8 h with TNF α (10 ng/ml, 4h)-stimulated BECs from control mice and iBEC-G153-KOs (data normalized to *Gapdh*, control set to 1, n=3-6). **C,D,** Gene expression was determined by qRT-PCR in HUVECs that were co-cultured with 4h growth medium-treated control and GPR153-kd hCASCs (n=6-8). Data are means \pm SEM; differences between genotypes were analyzed using two-way ANOVA and Sidak's post-hoc test. *, $P < 0.05$; **, $P < 0.01$; ***, $P < 0.001$; ****, $P < 0.0001$.

Taken together, GPR153-deficient ECs and SMCs show reduced production of pro-inflammatory mediators that have the capacity to modulate the activation state of neighbouring cells. In our transwell co-culture system, however, these differences were – except for reduced upregulation of *Vcam1* in maoSMC in the presence of GPR153 knockout BEC - not strong enough to induce prominent changes in the responder cells. This might be due to the relatively high distance of 1.27 mm between the transwell membrane and the lower compartment, as well as the relatively large dilution volume of 200 μ l; in addition, it is possible

that absence of direct contact plays a role. We hypothesize that *in vivo*, where ECs and SMCs are in close proximity and dilution is less relevant, the observed differences in mediator production might be more relevant. We attempted to address this experimentally, unfortunately we failed to harvest sufficient numbers of ECs from ligated carotid arteries of SM-specific KOs, and also the amount of SMCs harvested from CNS-vessels of BEC-specific KOs was not sufficient for analyses. However, we have included the new mRNA and protein data for BECs in Figs. 8D and 8E and describe them on page 9, lines 28-29), the IL-6 data for hCASMCs were included as Suppl. Fig. 2H and are described on page 6, line 24. Also, we included a statement addressing the potential interaction between ECs and SMCs in the discussion on page 13, lines 33-35:

“In the absence of GPR153, inflammatory activation is reduced both in ECs and SMCs, and this includes a reduced production of pro-inflammatory soluble mediators which might also affect the activation state of neighboring cells.”

3. If orphan receptor GPR153 facilitates pro-inflammatory and pro-proliferative gene expression in ECs and SMCs by controlling cellular cAMP levels. cAMP, the second messenger, could be controlled by many GPCRs. It is difficult for me to understand there is differences between GPR153 and other GPCRs controled cAMP.

This is certainly a crucial question, and we currently do not fully understand how GPR153 modulates cAMP levels. Our data clearly show that inactivation of GPR153 results in increased cAMP levels in all tested cell types (ECs, SMCs, HEKs), and our HEK data show that overexpression of GPR153 reduces cAMP levels. Based on these data we speculated that GPR153 is a $G_{i/o}$ -coupled receptor with a certain constitutive activity, since $G_{i/o}$ activity is expected to reduce cAMP levels through inhibition of adenylyl cyclases. However, GPR153-mediated suppression of cAMP was not sensitive to $G_{i/o}$ inhibitor pertussis toxin in any of our experiments, indicating that GPR153 modulates cAMP by other means. We therefore next hypothesized that GPR153 might interfere with G_{α_s} -mediated activation of adenylyl cyclases, and to test this hypothesis, we investigated whether GPR153-mediated suppression of cAMP was preserved after siRNA-mediated knockdown of *GNAS*, the gene encoding G_{α_s} . These data showed that in G_{α_s} -deficient HEK cells, knockdown of GPR153 no longer resulted in increased cAMP production, suggesting that GPR153 normally inhibits cAMP production in a G_{α_s} -dependent manner. In line with this notion, the suppressive effect of GPR153 overexpression on PGE₂-induced cAMP production was lost in G_{α_s} -deficient HEK cells.

Taken together, our data suggest that GPR153 is neither a typical $G_{i/o}$ -coupled receptor (since its effects are PTX insensitive) nor is it a typical G_s -coupled receptor (since this would have opposite effects on cAMP levels). Instead, GPR153 seems to regulates cAMP through a yet unknown mechanism that is $G_{\alpha_{i/o}}$ -independent, but G_s -dependent. This unique mechanism of action sets GPR153 apart from other GPCR and explains why it is able to even affect basal and agonist-induced cAMP levels. To make this point clearer, we have amended out statement on page 13 as follows:

“The mechanism by which GPR153 suppresses cAMP levels is currently not understood and probably differs from canonical G-protein signaling. A previous study showed that certain orphan GPCRs have constitutive activity towards $G_{\alpha_{i/o}}$ family G-proteins, resulting in inhibition of cAMP mediated effects.¹¹ GPR153 was not analyzed in that

particular study, so we tested whether GPR153-mediated reduction of cAMP was sensitive to the $G_{i/o}$ inhibitor PTX, but found this not to be the case. Also other studies did not find evidence for constitutive activity of GPR153 towards $G\alpha_{i1}$ or other G-proteins.^{12, 13} Instead, we found that GPR153-mediated suppression of cAMP levels depended on the presence of $G\alpha_s$, suggesting that GPR153 acts by interfering with $G\alpha_s$ -dependent adenylyl cyclase activation.

Reviewer 2

Vascular remodeling and inflammatory responses are important pathological processes upon various vascular damages. In this manuscript, the authors have shown that the expression of an orphan GPCR, GPR153 is upregulated in vascular SMCs and ECs in response to injuries. Then down-regulation (or deficiency) of its expression decreases proliferation and dedifferentiation of SMCs and inflammatory responses in ECs. The authors also have delineated the mechanism, in which GPR153 plays a role in pro-proliferative and pro-inflammatory gene expression. They have shown that increased cAMP levels in GPR153-deficient cells, leading to enhancing the extent of CREB phosphorylation that impedes proliferation and dedifferentiation of SMCs and inflammatory responses in ECs. Then the authors further have shown that loss of GPR153 expression leads to partial protection from neointima formation in the carotid artery ligation model and reduced severity of neuroinflammation and ischemic damage, which suggests that GPR153 may be a pathological target for parallel modulation of EC inflammation and SMC proliferation upon injuries. Overall, experiments and results presented in this manuscript were done in a relatively meticulous way and support the conclusion in the study, respectively. Furthermore, the present study will bring many attentions in the field of GPCRs and damage-promoted vascular remodeling and inflammatory responses. Nonetheless, there are several major and minor issues to be addressed in this form of the manuscript.

We thank the reviewer for the supportive comments and have addressed all major and minor points raised below.

Major

- 1. Throughout the manuscript, there are a few of descriptions that are confusing to easily follow. The authors are recommended to more explicitly describe them. For example:**
 - a. In the summary, lines 42-44, “Mechanistically, we show that GPR153 negatively regulates cellular cAMP levels, resulting in increased CREB phosphorylation, reduced YAP/TAZ levels, and diminished NF-κB activation in the absence of GPR153.” It would be better to be rephrased like “Mechanistically, we show that GPR153 negatively regulates cellular cAMP levels, and thus the absence of GPR153 leads to an increase in CREB phosphorylation, reduced YAP/TAZ levels, and diminished NF-κB activation”.**
 - b. The sentence on lines 264-266. One suggestion would be "Taken together, GPR153 regulates cellular cAMP levels in a Gi/o-independent inhibition of Gs activation", thereby reducing CREB-mediated signaling that inhibits YAP/TAZ and NF-κB-dependent signals".**
 - c. The sentence on lines 277-278 needs something like “in GPR153 knocked-down cells” at the end, and the sentence on lines 278-279, “upon GPR153 knocking-down” at the end.**
 - d. On the line-313, it needs to add “in GPR153-deficient BECs” after “.....significantly reduced expression of Vcam1”.**

We appreciate these suggestions very much and have implemented them all (page 1, lines 7-9; page 8, lines 17-19; page 8, lines 31-33; page 10, line 2).

- 2. In the experiment shown in Fig.6P, the authors pretreated cells with PTX for 3h. But the inhibitory effect of PTX often clearly appears after overnight pretreatment. Thus, it is recommended that the authors need to re-test the effect of PTX after overnight pretreatment, particularly for the**

case with the overexpressed receptor in this study before they draw the current conclusion. Although the authors performed a control experiment for PTX activity with CXCR4 stimulated with SDF1 α , the level of receptor expression and the coupling efficiency to Gi between GPR153 and CXCR4 may be different. Further, it seems not to be logical that the authors made the conclusion that "GPR153 normally inhibits cAMP production by interfering with Gas-dependent activation of adenylyl cyclases" through the experiment with Gs α -deficient cells as shown in Fig. 6Q and 6R.

This is a valid point, and we have repeated the experiment shown in Fig. 6P in cells that have been pre-treated with PTX for 16 hours. Also after prolonged PTX pretreatment, GPR153-induced suppression of cAMP levels was not altered (Fig. 4), suggesting that these effects are indeed G_{i/o}-independent.

Figure 4: Effect of PTX (100 ng/ml, 16h) on PGE₂ (1 μ M, 30 min)-induced cAMP production was determined by ELISA in EV- and GPR153-EP-transfected HEK cells (n=6). Data are means \pm SEM; differences between genotypes were analyzed using two-way ANOVA and Sidak's post-hoc test. EV, empty vector; PTX, pertussis toxin, GPR153-EP, expression plasmid encoding GPR153; ****, $P < 0.0001$.

We have added these data as Suppl. Fig. 5C and mention them on page 8, lines 9-10.

As to the conclusion from the experiment with G α_s -deficient cells in Fig. 6Q and 6R, we rephrased as follows: "We found that in G α_s -deficient HEK cells, knockdown of GPR153 no longer resulted in increased cAMP production (Fig. 6Q), and also the suppressive effect of GPR153 overexpression on cAMP production was lost in the absence of G α_s (Fig. 6R). Taken together, GPR153 reduces cellular cAMP levels through a yet unknown mechanism that is G $\alpha_{i/o}$ -independent, but G α_s -dependent. In the absence of GPR153, cAMP/CREB signaling is enhanced, resulting in inhibition of YAP/TAZ- and NF- κ B-dependent signals."

3. *On lines 282-284, it is described "Reduced expression of VCAM1 was confirmed on the protein level (Fig. 7M), whereas proliferation, here determined by EdU incorporation, was not significantly altered (Fig. 7N)." The authors need to briefly discuss what will be the reason why the proliferation was not significantly altered even though the expression of proliferation markers shown in Fig. 7L was substantially changed.*

It is indeed puzzling that despite reduced expression of transcripts such as PCNA or MKI67 we did not observe reduced EdU incorporation in HUVECs. Several factors may underlie this finding:

- A. Differences between RNA and protein level: Despite reduction on the mRNA level, protein levels of PCNA, MKI67 or CCNA2 might not be significantly reduced, for example because of compensatory changes in protein translation or stability.
- B. Normal function despite reduced protein levels due to protein reserve: In HeLa cells it was observed that a maximum of one-third of the total PCNA is tightly associated with the nucleus at peak S phase, suggesting that these cells maintain excess levels of PCNA.¹⁴ Moderate reduction of PCNA protein levels might therefore be without functional consequence.
- C. Normal function despite reduced protein levels due to altered posttranslational modification: PCNA, for example, exerts its functions through numerous protein interactions that can be regulated by post-translational modifications, including ubiquitination, NEDDylation, ISGylation, SUMOylation, acetylation and phosphorylation.¹⁵ Changes in these modification processes may compensate reduced expression of PCNA.
- D. HUVECs might have redundant pathways and compensatory mechanisms that maintain proliferation rates even if the functionality of PCNA, MKI67, or CCNA2 is compromised.

Options B-D are difficult to test, but we investigated option A, namely that reduced PCNA transcript levels did not result in altered protein levels. To do so, TNF α -treated control and GPR153 knockdown HUVECs were generated as before (see manuscript Figures 7L-N) and immunoblotting using antibodies directed against PCNA was performed. These analyses did not show reduction in PCNA levels in GPR153 knockdown HUVECs (Fig. 5A,B), suggesting that the observed reduction of PCNA transcripts by approximately 40% does not affect protein levels. The exact mechanisms underlying this compensation are currently not known.

Figure 5: A,B, Immunoblot analysis of lysates from control and GPR153 knockdown HUVECs 4 h after stimulation with 10 ng/ml TNF α : representative immunoblots (A; GAPDH as loading control) and densitometric analysis (B). Data are means \pm SEM; differences between genotypes were analyzed using unpaired t test.

Since we here addressed only PCNA, not the other genes, we have added a more general statement on page 11, line 34, to page 12, line 3: “In contrast to SMCs, GPR153-deficient ECs did not show reduced proliferation, even though expression of proliferation markers was

diminished on the mRNA level. The reasons for these differences are currently unknown, but EC-specific regulation of protein stability or posttranslational modification might be contributing factors.”

4. *It would be better if the authors more explicitly describe the conclusion from the adhesion/migration results observed as shown in Fig. 7O-R.*

Following the reviewer’s suggestion, we have added the following summary to the paragraph on GPR153 deficiency in HUVECs: “Taken together, GPR153 knockdown in HUVECs results in increased cAMP/CREB signaling and reduced activity of TAZ and NF- κ B pathways. Functional consequences are diminished inflammatory activation in TNF α -stimulated HUVECs, with consecutive reduction of leukocyte adhesion, leukocyte transmigration, and inflammation-induced hyperpermeability.” (page 9, lines 11-15)

5. *The authors described “In line with reduced inflammatory activity of the blood brain barrier,” on the line-315,”. However, it does not seem to be plausible to say this based on the observation shown in Fig. 8F where there is no statistically significant difference in the expression of inflammatory markers except Vcam1 between control and iBEC-G153-KO cells.*

This is a valid point, and we performed additional EAE experiments to clarify whether the reduced expression of inflammatory markers observed in Fig. 8F is sufficiently strong to affect immune cell infiltration (and, consequently, EAE susceptibility) in iBEC-G153-KOs. To do so, we determined numbers and identity of spinal cord-infiltrating leukocytes by flow cytometry at the onset of EAE (day 14) and found that total leukocytes (CD45⁺), monocytes (CD45⁺, CD11b⁺, Ly6C^{hi}) and neutrophils (CD45⁺, CD11b⁺, Ly6G⁺) were significantly reduced in iBEC-G153-KOs (Figs. 6A,B). To investigate whether reduced leukocyte infiltration was associated with reduced demyelination of the white matter, we performed luxol fast blue staining of white matter in lumbar spinal cords of mice harvested 28 days after immunization. These data showed that KO mice have significantly reduced demyelination in the spinal cord (Fig. 6C,D).

Figure 6: A,B, Flow cytometric analysis of spinal cord-infiltrating leukocytes on day 14: Absolute numbers of CD45⁺ leukocytes (All), CD45⁺,CD11b⁺,Ly6G⁺ neutrophils (Neutro), CD45⁺,CD11b⁺,Ly6C^{hi} monocytes (Mono), CD11b⁺, CD45^{hi}, Ly6C⁺ macrophages (Macroph), CD45⁺,CD19⁺,CD4⁺ T cells (CD4 T), CD45⁺,CD19⁺,CD8⁺ T cells (CD8 T) (n=9-10). **C,D**, Quantification of demyelination by luxol fast blue staining on day 28 after induction of EAE: Exemplary photomicrographs (C) and statistical evaluation (D) (18 and 21 sections from 6 and 7 mice, respectively). Data are means ± SEM; differences between genotypes were analyzed using two-way ANOVA with Sidak's post hoc test (B) or unpaired two-sided t test (D). **, $P < 0.01$; ****, $P < 0.0001$.

We have added the new data for immune cell infiltration as Fig. 8I to the revised manuscript and describe them on page 10, lines 4-5. The data on demyelination on day 28 are included as Fig. 8K and L and are describe them on page 10, lines 8-9.

Minor

6. *In Fig. 3J, it looks like there are some decreases in contractile responses of iSM-G153-KO mice compared to those in control animals unlike what the authors described. If so, it would be better to say "Taken together, inactivation of GPR153 in SMCs does not substantially affect basic vascular function, on the line-177.*

The text has been changed as suggested (page 5, line 22)

- 7. In Fig. 5, the authors may need to briefly describe what was the reason why they examined hCASMCs more details before AoSMCs to understand how GPR153 regulates SMC function since the previous results were obtained from AoSMCs.**

This is a valid question, though the answer is rather practical in nature: The efficiency of siRNA-mediated knockdown in hCASMCs was slightly more efficient than esiRNA-mediated knockdown in hAoSMC, which made us choose hCASMCs for further analyses. We have added a corresponding statement on page 6, line 6-8.

- 8. On the line-207, "select" must be "selected". Further, is there any rational why the expression of the selected target genes of CREB (Fig. 5J) and NF-κB (Fig. 5O) is changed while the others are not.**

The text has been changed as suggested (page 6, line 19).

The question why only some of the CREB, YAP/TAZ, or NF-κB target genes are affected by GPR153 knockdown is indeed interesting. A likely explanation lies in the fact that most genes are not regulated by only one transcription factor, but by a plethora of transcriptional regulators and epigenetic mechanisms. Whether changes in one particular transcription factor have significant effects on target gene expression will therefore depend on the number of alternative regulators available in the respective cell type. *Tnf* expression, for example, is not only regulated by NF-κB, but also by transcription factors of the STAT and IRF families, as well as LPS-induced TNF-α factor LITAF¹⁶. We therefore hypothesize that in those genes that are not affected by GPR153 knockdown other regulatory mechanisms can compensate the effects of enhanced CREB activity or reduced YAP/TAZ or NF-κB activity. However, since we currently do not have hard data to explain the observed differences, and also because the discussion is already quite lengthy, we prefer not to add a paragraph addressing this topic.

- 9. On the line-210, "NF-κB target genes cells" must be "selected NF-κB target gene".**

The text has been changed as suggested (page 6, line 23).

- 10. In Fig. 5P, the authors have shown the reduced TAZ level, how about the extent of YAP as shown in the serum-starvation condition (Fig. 5A and C).**

Following the reviewer's suggestion, we have analyzed phosphorylated YAP and total YAP in hCASMCs in the presence of growth medium and found similar changes as in the serum-starved state, namely increased levels of phosphorylated YAP and reduced levels of total YAP in GPR153 knockdown cells (Fig. 7A-C).

Figure 7: A-C, Control and GPR153 knockdown hCASMCs were starved for 48 hours and then exposed to growth medium for 15 min, followed by immunoblotting (same procedure as in main Figure 5K-P, GAPDH as loading control, n=6). Data are means \pm SEM; differences between genotypes were analyzed using unpaired two-sided t test. *, $P < 0.05$; **, $P < 0.01$.

We have added these new data to Supplemental Figure 2I-K and describe them on page 6, lines 24-25.

11. On the line-214, it would be better to add "in spite of no statistical significance" after "We found that nuclear MRTF-A localization was in trend increased in GPR153-deficient cells in the starved state,"

We changed the sentence as follows: "Nuclear MRTF-A localization showed a non-significant increase in GPR153-deficient cells in the starved state, whereas no differences were observed in the presence of serum (Fig. 5Q,R)." (page 6, lines 29-31)

12. On the line-219, it would be better to add ""in selected target genes" at the end of the sentence.

The text has been changed as suggested (page 7, line 2).

13. In the figure legend for Fig. 6K, it would be clearer to change "fold changes" to "PGE2-induced fold changes over basal".

The legend has been changed as suggested.

14. On the line-259, "cAMP" would be better with "the cAMP level".

The text has been changed as suggested (page 8, line 12).

15. In the figure legend for Fig. 7A. it is described as "data reanalyzed from71". the reference-71 must be the -23 as described in the text.

That is true, and we corrected the mistake.

16. In Fig. 8C, the authors may need to describe what would be the reason while *Vcam1* expression was impaired, *Icam1* expression did not altered in GPR153-deficient BECs compared to the control cells?

It is true that *ICAM1* reduction did not reach significance in the data set shown in the previous version of Fig. 8C ($p=0.2$). This is indeed surprising, since we observed a clear reduction of both *ICAM1* and *VCAM1* in GPR153 knockdown HUVECs (Fig. 8A, same as manuscript Fig. 7L), and also because an initial analysis in BECs showed a reduction of both genes (Fig. 8B). We have now added the previously not included data points shown in Fig. 8B to the analysis in manuscript Fig. 8C, which now reaches significance (see Fig. 8C below). We have also investigated *ICAM1* on the protein level and also here reduced expression was observed (Fig. 8D). Both the updated mRNA data and the new protein data have been included in Figs. 8C and 8F of the revised manuscript, and they are described on page 9, lines 26 and 30.

Figure 8: **A**, Gene expression was determined by qRT-PCR in TNF α -treated control and GPR153 knockdown HUVECs (data normalized to *GAPDH*, control set to 1, $n=8$) (same as manuscript Figure 7L). **B,C**, Gene expression was determined by qRT-PCR in TNF α (10 ng/ml, 2h)-stimulated BECs cultured from control mice and iBEC-G153-KOs (data normalized to *Gapdh*, control set to 1): B, Data from a previously unused data set ($n=3$ and 5); C, new version of Figure 8C after addition of data shown in B ($n=14-19$). **D**, Expression of *ICAM1* and *VCAM1* was determined by immunoblotting in lysates of TNF α (10 ng/ml)-stimulated BECs from control and iBEC-G153-KO mice (representative of 2 experiments).

Data are means \pm SEM; comparisons between genotypes were performed using two-way ANOVA with Sidak's multiple comparison test. *, $P < 0.05$; **, $P < 0.01$; ***, $P < 0.001$.

17. On the line-314, it would be better to add "in spite of no statistical significance" at end of the sentence for better description of the corresponding result shown in Fig. 8F

The text has been changed.

Reviewer #3 (Remarks to the Author):

In this data-enriched submission, the authors described their investigation of the functions of the orphan receptor GPR153 in vascular cells. They generated smooth muscle cell-specific or endothelial cell-specific GPR153 conditional knockout mouse lines, which is a strength of the study. They conclude that GPR153 facilitates pro-inflammatory and pro-proliferative vascular damage responses by regulating vascular smooth muscle cells and endothelial cells, particularly the cAMP/CREB signaling. Unfortunately, the manuscript is a collection of descriptive datasets generated using various cell cultures and animal models without focused scientific hypotheses being tested. For instance, the last figure describes three disease models (EAE, stroke, and TBI); however, the data are too preliminary to draw meaningful conclusions. The manuscript is extremely difficult to follow, as the logical flow is unclear to the readers.

We thank the reviewer for the helpful criticism and hope that the experiments performed in response to the specific points help to address the reviewer's concerns.

Specific critiques:

1. The first half of the study (Figs. 1-5) investigated the role of GPR153 in vascular smooth muscles using cell cultures and an in vivo carotid artery occlusion model. The main finding in cell cultures was that GPR153 knockdown inhibits the proliferation of smooth muscle cells under basic conditions or in the presence of growth factors. The functional significance of such an effect is unclear.

In healthy mice, the functional significance of reduced proliferation of GPR153-deficient SMCs seems indeed limited, most likely due to the low proliferative activity of healthy, quiescent SMCs. However, after induction of vascular damage in the carotid ligation model, we observed a significant reduction of neointima formation, and we hypothesize that reduced proliferation is a major factor underlying this finding. In line with this notion, qRT-PCR analyses in the media of carotid arteries harvested on day 7 after ligation showed that proliferation marker *Mki67* was upregulated in control samples, but not in KO samples (Fig. 9A). To further support the notion of reduced proliferation in GPR153-deficient SMCs in the neointima model, we determined EdU incorporation in the media of mice that had been subjected to carotid artery ligation. These analyses showed that on day 14 after ligation, iSM-G153-KOs showed significantly fewer EdU-positive cells in the α SMA-positive media than control mice, indicative of reduced proliferation *in vivo* (Fig. 9B,C).

Figure 9: A, Expression of proliferation marker *Mki67* was determined by qRT-PCR in the cleaned media of ligated carotid arteries (Ligation +) and non-ligated contralateral vessels (Ligation -) harvested on d7 after neointima (NI) induction from control mice (Con) and iSM-G153-KOs (KO) (data normalized to *Gapdh*, non-ligated control set to 1, n=6-8). **B,C,** Carotid arteries harvested on day 14 after ligation from mice that had received EdU injections every second day from day 6-14 were cryosectioned and stained for α SMA and EdU: Exemplary microphotographs (B, hatched yellow line indicates media borders as judged by α SMA staining) and statistical evaluation (C, n=9).

Data are means \pm SEM; differences between genotypes were analyzed using two-way ANOVA with Sidak's post hoc test (A) or unpaired two-sided t test (C). **, $P < 0.01$; ***, $P < 0.001$.

Together, these data show that reduced proliferation of GPR153-deficient SMCs is not only an *in vitro* phenomenon, but also observed *in vivo* in the carotid artery damage model, where it contributes to the protective effect of SMC-GPR153 deficiency. We have added these data as Figures 4F and 4G to the revised manuscript and describe them on page 5, line 31-33.

They later studied the signaling pathway under GPR153 regulation using human coronary artery smooth muscle cells. They found that GPR153 knockdown upregulates CREB, YAP/TAZ, and NF- κ B signaling. Again, the functional significance of this pathway is unclear. No follow-up experiments were performed to determine the functional consequence of manipulating this signaling pathway.

We are not quite sure whether "this signaling pathway" here refers to GPR153 itself or to the signaling cascades regulated by GPR153, namely CREB, YAP/TAZ, and NF- κ B. Since the functional consequences of GPR153 deletion in SMCs are addressed in Figures 2 (*in vitro*) and Figure 3 and 4 (*in vivo*), we assume that the latter interpretation is correct and would like to respond as follows:

It is true that we so far did not perform experiments to address the functional consequences of manipulating CREB/YAP/TAZ and NF- κ B in SMCs. The main reason for this was that there is already an extensive body of literature addressing the role of these signaling pathways in SMCs, and we attempted to summarize these studies in the discussion on page 11, lines 12-24, page 12, lines 4-14, and page 12, lines 17-24.

However, we agree that it is interesting to experimentally test whether the above-mentioned changes on the signaling level indeed underlie functional consequences such as

reduced proliferation in GPR153-deficient SMC. In our HEK studies, we showed that stimulation of adenylyl cyclases by forskolin (FSK) mimics the consequences of GPR153 deficiency on the signaling level: CREB phosphorylation is increased, while TAZ levels and NF- κ B activation are reduced. FSK treatment might therefore be used as a tool to generate a “GPR153 knockdown-like” signaling phenotype, and allows us to test whether these alterations indeed affect proliferation in SMC. We therefore investigated whether a) FSK-mediated stimulation of cAMP production/CREB phosphorylation resulted also in hCASMCs in reduced levels of TAZ and NF- κ B activation and b) whether this was associated with reduced proliferation. Immunoblotting analyses showed that FSK-induced CREB phosphorylation indeed preceded a significant reduction in total TAZ levels (Fig. 10A-C), and also NF- κ B p65 phosphorylation was time-dependently reduced by FSK (Fig. 10D,E). We next tested whether FSK treatment resulted in reduced proliferation, and found a significant reduction of cell growth rate in FSK-treated control cells (Fig. 10F, left). Interestingly, FSK-mediated reduction of growth rate was no longer observed in GPR153 knockdown SMCs (Fig. 10F, right), indicating that their basal elevation of cAMP/CREB signaling already resulted in maximal CREB-dependent suppression of proliferation.

Figure 10: A-E, Immunoblotting of lysates from hCASMCs in the basal state or after stimulation with 1 μ M FSK for the indicated times: representative blots (A,D, GAPDH as loading control) and densitometric analysis of signal strength (B,C,E; n=3-4). **F**, Effect of 1 μ M FSK on cell number expansion in control and GPR153 knockdown hCASMCs (n=3). Data are means \pm SEM; differences were analyzed using one-way ANOVA with Dunnett’s multiple comparisons test (B,C,E) or unpaired two-sided t tests corrected for multiple testing with the two-stage step-up method (Benjamini, Krieger, and Yekutieli) (F). *, $P < 0.05$; **, $P < 0.01$; ***, $P < 0.001$.

We have added these new data as Supplemental Figures 4A-F to the revised manuscript and describe them on 7, line 29-32.

2. (A) They generated SMC-specific GPR153 cKO mice and performed the carotid artery ligation model. They found that GPR153 cKO inhibits SMC proliferation, differentiation, and neointima formation. However, the data are too preliminary to conclude, as the functional impact of GPR153 is unclear, and the signaling mechanism is unknown.

As mentioned by the reviewer, our data not only show that neointima formation is reduced in iSM-G153-KOs, we also provide clear evidence that SMCs harvested from ligated carotids of KO mice show reduced downregulation of contractile marker genes and hardly any upregulation of proliferation markers (Figures 4E and 4H-K of the revised manuscript). In addition, we now also provide data showing reduced EdU staining in vessel sections harvested on day 14 after ligation (Fig. 11A,B; see also our response to question 1 above). We have integrated these new data in Figures 4F and 4G of the revised manuscript and believe that this makes the functional impact of GPR153 on SMC proliferation in the NI model more evident

With respect to the mechanisms underlying reduced proliferation and dedifferentiation in iSM-G153-KOs, we attempted to study CREB phosphorylation in SMCs isolated by FACS from mice subjected to the neointima model. Unfortunately, it was not possible to isolate enough SMCs from the 2 mm carotid artery segment that shows neointima formation to allow analysis by immunoblotting. We therefore assessed CREB phosphorylation by immunofluorescence staining in ligated carotid arteries harvested from control and iSM-G153-KOs of day 7 after ligation. These analyses showed that CREB phosphorylation was indeed significantly elevated in carotid arteries of KO mice (Figs. 11C,D).

Figure 11: A, B, Carotid arteries harvested on day 14 after ligation from mice that had received EdU injections every second day from day 6-14 were cryosectioned and stained for α SMA and EdU: Exemplary microphotographs (B, hatched yellow line indicates media borders as judged by α SMA stain) and statistical evaluation (C, n=9). **C, D,** Carotid arteries harvested on day 7 after ligation from control and iSM-G153-KOs were cryosectioned and immunostained with antibodies directed against α SMA and pCREB: Exemplary microphotographs (C, hatched yellow lines indicate media borders as judged by α SMA staining) and statistical evaluation of pCREB signal within the α SMA-positive area (D, n=9).

Data are means \pm SEM; differences between genotypes were analyzed using unpaired two-sided t test. *, $P < 0.05$; ***, $P < 0.001$.

Unfortunately, even though we tried four different antibodies for YAP/TAZ (p-YAP Ser127 (CST 4911), YAP/TAZ (CST 8418), TAZ (ab242313), TAZ (sigma HPA007415)), none of them worked reliably in immunofluorescence staining. However, immunoblotting in SMCs cultured from control and KO mouse vessels showed that also in primary murine SMCs elevated CREB phosphorylation is associated with reduced TAZ levels (Fig. 12A-D).

Figure 12: A-D, Immunoblotting of serum-starved murine aortic SMCs cultured from control and iSM-G153-KOs (GAPDH as loading control, n=6). Data are means \pm SEM; differences between genotypes were analyzed using unpaired two-sided t test. *, $P < 0.05$; **, $P < 0.01$.

Taken together, we now provide additional evidence that inactivation of GPR153 reduces proliferation not only in cultured human SMC, but also in the carotid media of iSM-G153-KOs subjected to the ligation model. Since proliferation is a major factor underlying neointima formation,^{17, 18} we propose that this – in addition to reduced dedifferentiation – importantly contributes to the reduced neointima burden in iSM-G153-KOs. These new data are shown in Figures 4F,G and described on page 5, line 31-33.

Regarding the signaling mechanism, we now show that increased CREB phosphorylation, the putative driver of altered YAP/TAZ levels and NF- κ B phosphorylation (see Fig. 5 of revised manuscript), is not only observed in cultured human SMC, but also in the media of ligated carotid arteries of iSM-G153-KOs. These new data have been added to the revised manuscript as Figure 5W and Suppl. Fig. 3I and are described on page 6, line 33-35.

(B) Using *Myh11-CreERT2* mice to generate SMC-specific GPR153cKO mice has several caveats that may affect data interpretation. First, the *Myh11-CreERT2* transgene is located on the Y chromosome, which means it can only be used to study male mice. However, sex dimorphism of vascular structure and function has been increasingly recognized.

It is true that *Myh11-CreERT2* mice can only be used to study male mice, and sex dimorphism can therefore not be addressed in the SMC model *in vivo*. However, our *in vitro* data in hCASMCs were done in cells from both male and female donors, and the phenotypes of GPR153-deficiency were present in cells from both sexes. Furthermore, our studies in iBEC-G153-KOs allowed to investigate both male and female mice, and also here, no difference between the sexes were observed. Taken together, though it is true that *Myh11-CreERT2* allows only analysis in male mice, the described changes in cAMP/CREB/YAP/TAZ/NF- κ B signaling have been observed in cells from both male and female human and murine donors and no differences between the sexes were observed. Information regarding the sex of

experimental animals are presented on page 17, lines 1-2 (mice), page 19, lines 17 (maoSMCs) and 31 (BECs), page 18, line 26 (human SMC) and 32 (HUVEC).

(C) Second, the *Myh11-CreERT2* transgene can recombine in vascular and non-vascular smooth muscles. Thus, the effects of *GPR153* cKO on non-vascular smooth muscles should be considered.

This is a valid point, the *Myh11-CreERT2* line indeed recombines also in non-vascular SMC. We carefully monitored our mice for signs of gastrointestinal/urogenital dysfunction after tamoxifen treatment, but did not observe obvious differences. For example, body weight was not altered 14 days after tamoxifen induction (Fig. 13A), and also faeces production was normal (Fig. 13B). Post-mortem inspection of the gastrointestinal and urogenital tract did not reveal any abnormalities, and gut length and structure appeared normal (Fig. 13C-E). To assess micturition, we performed the void spot assay, in which a mouse is placed in an enclosure lined by filter paper and the resulting urine spot pattern is quantified.¹⁹ This assay did not reveal differences with respect to the pattern and total area of urine released within 4 hours (Fig. 13F,G).

Figure 13: **A**, Body weight of control mice and iSM-G153-KOs 14 days after end of tamoxifen induction. **B**, Faeces production per 24 h in control mice and iSM-G153-KOs 14 days after end of tamoxifen induction. **C-E**, Morphometric analysis of intestines from control mice and iSM-G153-KOs 14 days after end of tamoxifen induction: exemplary photographs showing straightened intestines of both genotypes (C) as well as statistical analysis of small intestine length (D) and colon length (E). **F,G**, Exemplary photomicrographs (F) and statistical evaluation (G) of the urine release pattern within 4 h (n=5). Data are means \pm SEM; differences between genotypes were analyzed using unpaired two-sided t test.

Taken together, we did not observe evidence for major abnormalities in the function of non-vascular SMCs *in vivo*, but more detailed analyses, especially under pathophysiologically

relevant conditions, might of course reveal differences between control mice and iSM-G153-KOs. However, since the focus of this study is on GPR153 function in vascular cells, we here prefer not to follow up on this interesting question.

3. Figure 6 shows the effects of GPR153 knockdown on cellular cAMP levels and pCREB/TAZ signaling. These experiments were done using HEK cells. The significance of the data is unclear, as HEK cells are not vascular cells.

We agree that we should have made the rationale for using HEK cells in Figure 6 clearer. There were actually three reasons why we decided to complement our data in vascular cells by studies in HEK cells: Firstly, HEK cells are – in contrast to SMCs – easy to transfect with plasmids and therefore allowed us to study the consequences of GPR153 overexpression. Secondly, the transfectability of HEK cells allowed the use of transfection-based readout systems, for example the cAMP GloSensor system. Thirdly, we did not consider it a disadvantage that HEK cells are not vascular, but rather an advantage, because it indicates that the observed consequences of GPR153 deficiency are not vascular-specific, but of a more global nature. However, we made sure all key findings in GPR153 knockdown HEK cells (elevation of cAMP/pCREB and reduced TAZ/NF- κ B activation) are also shown in both SMCs and ECs, and also FSK-dependent effects on TAZ/NF- κ B have now been reproduced in SMCs (see Fig. 10 of this letter). To better explain the rationale for using HEK cells, we added the following statement to the revised version of the manuscript (page 7, line 10-14):” To test this hypothesis, we studied GPR153 effects in human embryonic kidney (HEK) 293T cells, which allow not only siRNA-mediated knockdown, but also plasmid-based overexpression of GPR153 (Fig. 6A). In addition, plasmid-based reporter systems such as the cAMP GloSensor can be efficiently used in HEK cells”

4. They generated EC-specific GPR153 cKO mice and performed injury models using both GPR153 cKO and WT control mice, including the EAE model and a focal brain stroke model (Fig. 8). However, the data are too preliminary to conclude. In the EAE study, essential pathological data are lacking to demonstrate the histological basis for GPR153 cKO-afforded protection.

Following the reviewer’s suggestion, we performed additional experiments to characterize the immunological and histopathological consequences of BEC-specific GPR153 deficiency in the EAE model. To corroborate the notion that reduced EAE scores in KOs are due to reduced leukocyte infiltration into the spinal cord, we repeated the experiment in a new batch of mice and analysed leukocyte infiltration at the onset of disease, in this case at day 14 after immunization (Fig. 14A). A FACS-based analysis showed that the total number of infiltrating leukocytes was significantly reduced in iBEC-G153-KOs, particularly the monocyte population (CD45⁺, CD11b⁺, Ly6C^{hi}) and the neutrophil population (CD45⁺, CD11b⁺, Ly6G⁺) (Figs. 14B).

To investigate whether reduced leukocyte infiltration was associated with reduced demyelination of the white matter, we performed luxol fast blue stainings in lumbar spinal cords of mice harvested 28 days after immunization. These data showed significantly reduced demyelination in the spinal cords of KO mice (Fig. 14C,D).

Figure 14: A,B, Flow cytometric analysis of spinal cord-infiltrating leukocytes on day 14: Absolute numbers of CD45⁺ leukocytes (All), CD45⁺,CD11b⁺,Ly6G⁺ neutrophils (Neutro), CD45⁺,CD11b⁺,Ly6C^{hi} monocytes (Mono), CD11b⁺, CD45^{hi}, Ly6C⁻ macrophages (Macroph), CD45⁺,CD19⁻,CD4⁺ T cells (CD4 T), CD45⁺,CD19⁻,CD8⁺ T cells (CD8 T) (n=9-10). **C,D,** Quantification of demyelination by luxol fast blue staining on day 28 after induction of EAE: Exemplary photomicrographs (C) and statistical evaluation (D) (18 and 21 sections from 6 and 7 mice, respectively). Data are means \pm SEM; differences between genotypes were analyzed using two-way ANOVA with Sidak's post hoc test (B) or unpaired two-sided t test (D). **, $P < 0.01$; ****, $P < 0.0001$.

We have added the new data regarding immune cell infiltration in the EAE model as Fig. 8I to the revised manuscript and describe them on page 10, line 4-5. The data on demyelination on day 28 are included as Fig. 8K and L and are described on page 10, lines 8-9.

The stroke outcome assessment is limited to the acute stage (48 hours) and lacks a dedicated assessment of neurological functions.

As suggested by the reviewer, we repeated the stroke model to include additional behavioral tests on day 2 and to add a later analysis time point for infarct size. Also in this new group, increases in the latency to move were less prominent in iBEC-G153-KOs than in control mice (Fig. 15A). In the corner test, in which mice with greater stroke volumes are more likely to turn via the unaffected (in this case right) limb, we found that KO mice were less biased towards

the right side than controls (Fig. 15B). In the tape test, which determines the time until an adhesive sticker attached to the affected hind paw is sensed and removed, KO mice performed significantly better (Fig. 15C,D). Also in this group, improved performance in behavioral testing was associated with reduced infarct volumes, in this case analyzed at a later time point, namely 5 days after MCAO (Figs. 15E)

Figure 15: A-D, Behavioral testing before (d0) and 2 days (d2) after MCAO: Latency to move one body length (A), number of right turns in the corner test (B), and latency to sense and remove an adhesive sticker in the tape test (C,D) (n=6-7). **E**, Histological analysis of infarct volume on day 5 (n=6-7).

Data are means \pm SEM; comparisons were performed using two-way ANOVA with Sidak's multiple comparison test (A-D) or unpaired two-sided t test (E). *, $P < 0.05$; **, $P < 0.01$; ***, $P < 0.001$.

The additional behavioral testing in the MCAO is shown in Fig. 9A-C (old Figure 8 has been split into Fig. 8 (*in vitro* and EAE) and Fig. 9 (MCAO), and are reported on page 10, lines 17-21. Infarct sizes on day 5 are shown in Fig. 9G and are described on page 10, lines 23-24.

The mechanisms by which endothelial GPR153 activation leads to neuronal injury in EAE or stroke are not investigated in the current study.

Especially after adding the new data shown in Figure 14 above, we believe that we now provide good evidence that altered CREB/TAZ/NF- κ B signaling in GPR153-deficient ECs is associated with reduced inflammatory activation and diminished expression of leukocyte adhesion molecules, for details please revisit Figures 7L-M for HUVECs, Figures 8C-F for cultured BECs, Figure 8H for sorted spinal cord ECs in EAE, and Figures 9H-I for brain sections in MCAO. This, in turn, results in reduced leukocyte adhesion and transmigration as shown in Figures 7O-Q (HUVEC), Figure 8G (cultured BEC), Figure 8I (spinal cord infiltration EAE), and Figures 9J-K (brain sections MCAO). Immune cell infiltration into brain and spinal cord is not only the driver of neuroinflammation such as EAE, it is also well known to contribute to the pathogenesis of ischemic brain damage: Neutrophils and other immune cells are recruited from the blood stream within hours of stroke onset and exacerbate tissue damage through the release of reactive oxygen species, proteolytic enzymes, and pro-inflammatory cytokines.²⁰ We therefore believe that the above-mentioned differences in EC activation and consecutive leukocyte recruitment are the main factors underlying the protective effects of GPR153 deficiency in both models.

References

1. Mack CP. Signaling mechanisms that regulate smooth muscle cell differentiation. *Arterioscler Thromb Vasc Biol.* 2011;31:1495-505.
2. Kirchmer MN, Franco A, Albasanz-Puig A, Murray J, Yagi M, Gao L, Dong ZM and Wijelath ES. Modulation of vascular smooth muscle cell phenotype by STAT-1 and STAT-3. *Atherosclerosis.* 2014;234:169-75.
3. Yoshida T and Hayashi M. Role of Kruppel-like factor 4 and its binding proteins in vascular disease. *J Atheroscler Thromb.* 2014;21:402-13.
4. Li M, Qian M, Kyler K and Xu J. Endothelial-Vascular Smooth Muscle Cells Interactions in Atherosclerosis. *Front Cardiovasc Med.* 2018;5:151.
5. Lilly B. We have contact: endothelial cell-smooth muscle cell interactions. *Physiology (Bethesda).* 2014;29:234-41.
6. Mendez-Barbero N, Gutierrez-Munoz C and Blanco-Colio LM. Cellular Crosstalk between Endothelial and Smooth Muscle Cells in Vascular Wall Remodeling. *International journal of molecular sciences.* 2021;22.
7. Viedt C, Vogel J, Athanasiou T, Shen W, Orth SR, Kubler W and Kreuzer J. Monocyte chemoattractant protein-1 induces proliferation and interleukin-6 production in human smooth muscle cells by differential activation of nuclear factor-kappaB and activator protein-1. *Arterioscler Thromb Vasc Biol.* 2002;22:914-20.
8. Kurozumi A, Nakano K, Yamagata K, Okada Y, Nakayamada S and Tanaka Y. IL-6 and sIL-6R induces STAT3-dependent differentiation of human VSMCs into osteoblast-like cells through JMJD2B-mediated histone demethylation of RUNX2. *Bone.* 2019;124:53-61.
9. Wettschureck N, Strilic B and Offermanns S. Passing the Vascular Barrier: Endothelial Signaling Processes Controlling Extravasation. *Physiol Rev.* 2019;99:1467-1525.
10. Theofilis P, Sagris M, Oikonomou E, Antonopoulos AS, Siasos G, Tsioufis C and Tousoulis D. Inflammatory Mechanisms Contributing to Endothelial Dysfunction. *Biomedicines.* 2021;9.
11. Martin AL, Steurer MA and Aronstam RS. Constitutive Activity among Orphan Class-A G Protein Coupled Receptors. *PLoS One.* 2015;10:e0138463.
12. Lu S, Jang W, Inoue A and Lambert NA. Constitutive G protein coupling profiles of understudied orphan GPCRs. *PLoS One.* 2021;16:e0247743.
13. Zeghal M, Laroche G, Freitas JD, Wang R and Giguere PM. Profiling of basal and ligand-dependent GPCR activities by means of a polyvalent cell-based high-throughput platform. *Nat Commun.* 2023;14:3684.
14. Morris GF and Mathews MB. Regulation of proliferating cell nuclear antigen during the cell cycle. *J Biol Chem.* 1989;264:13856-64.
15. Sogaard CK and Otterlei M. Targeting proliferating cell nuclear antigen (PCNA) for cancer therapy. *Adv Pharmacol.* 2024;100:209-246.
16. Falvo JV, Tsytsykova AV and Goldfeld AE. Transcriptional control of the TNF gene. *Curr Dir Autoimmun.* 2010;11:27-60.
17. Bennett MR, Sinha S and Owens GK. Vascular Smooth Muscle Cells in Atherosclerosis. *Circ Res.* 2016;118:692-702.
18. Cao G, Xuan X, Hu J, Zhang R, Jin H and Dong H. How vascular smooth muscle cell phenotype switching contributes to vascular disease. *Cell Commun Signal.* 2022;20:180.
19. Hill WG, Zeidel ML, Bjorling DE and Vezina CM. Void spot assay: recommendations on the use of a simple micturition assay for mice. *Am J Physiol Renal Physiol.* 2018;315:F1422-F1429.
20. Denorme F, Rustad JL and Campbell RA. Brothers in arms: platelets and neutrophils in ischemic stroke. *Curr Opin Hematol.* 2021;28:301-307.

NCOMMS-24-49753B: Response to Remaining Reviewer comments.

Reviewer #2:

In this revised manuscript, the authors have addressed well all the concerns raised from the original version except one as below.

In Fig.6 to address the main concern No. 5 in the rebuttal letter, while the data on panels B, C and D are well depicted, it is hard to understand the purpose of the data on panel A. First, there must be no statistical significance between the samples obtained from Control and iBEC-G153-KO.

Panel A in rebuttal letter Fig. 6 shows the neurological scores in the repetition experiment that was performed to determine leukocyte infiltration into the spinal cord at an early time point of EAE development, in this case 14 days after immunization. Due to the high variation there is at this early time point indeed no significant difference with respect to neurological score, but immune cell infiltration is clearly reduced in KO mice.

The second is how this data is different from the one shown in Fig. 8J in the revised manuscript.

The data set in Fig. 8J shows a completely independent experimental group, and this group was observed for 28 days after immunization.

The third is I have realized that there is no description how the authors obtained "neurological score" in the manuscript.

As to the description of the neurological score, the following information is included in the methods section on page 17, line 1-2: "From day 7 on, animals were monitored daily until day 28 and clinical scoring of EAE was conducted as follows: 0, no clinical disease; 1, limp tail; 2, impaired righting reflex and gait; 3, partial hind limb paralysis; 4, tetraparesis; 5, dead"

Reviewer #4:

This study may offer some fundamental advances in terms of mechanisms. However, the parts that try to apply these pathways to disease models are somewhat diluted. The effects on SMC proliferation are clear and exciting. So, would it have been nice to focus on this and see how this pathway affects some type of clinical disease like carotid stenoses or coronary artery narrowing? The effects on endothelial cells are also exciting if it is true that GPR153 mediates NFkB inflammation, upregulates adhesion molecules and then promotes infiltration of damaging immune cells into the CNS. But if so, would it be nice to just pick stroke and go deeper?

- a. For example, what happens to GPR153 and some of the downstream signals in the stroke model itself?*
- b. If GPR153 affects cAMP, what happens to vasoregulation and cerebral blood flow in normal or ischemic conditions?*
- c. The cell culture studies refer to permeability. What happens to the BBB after focal cerebral ischemia?*
- d. The authors show that VCAM1 is decreased and infiltrated immune cells are lower. Is there some spatial correlation of these two events with the GPR153 signals?*

- e. *If there are effects on demyelination in the EAE model, should one also see decreased immune cells in the corpus callosum after focal cerebral ischemia along with improved white matter integrity?*
- f. *Finally, in the stroke field, it is somewhat common to perform behavior tests for at least 1-2 weeks post-stroke. Would this approach be useful here?*

We thank the reviewer for the thoughtful comments and suggestions. These are all very interesting and important issues, and we very much would like to address the role of GPR153 in vasoregulation and vascular permeability in more detail. Also the spatial distribution of immune cell infiltration is a relevant question for future analyses, and we agree that more in-depth studies in other disease models and conditions are needed to fully characterize the role of GPR153 in vascular damage. To acknowledge the point we have added the following sentence to the discussion:

“However, our findings are currently limited to the murine models of ligation-induced neointima formation, EAE and ischemic stroke, and further studies are required to determine the relevance of GPR153 in other vascular damage models.”

Vascular remodeling and inflammatory responses are important pathological processes upon various vascular damages. In this manuscript, the authors have shown that the expression of an orphan GPCR, GPR153 is upregulated in vascular SMCs and ECs in response to injuries. Then down-regulation (or deficiency) of its expression decreases proliferation and dedifferentiation of SMCs and inflammatory responses in ECs. The authors also have delineated the mechanism, in which GPR153 plays a role in pro-proliferative and pro-inflammatory gene expression. They have shown that increased cAMP levels in GPR153-deficient cells, leading to enhancing the extent of CREB phosphorylation that impedes proliferation and dedifferentiation of SMCs and inflammatory responses in ECs. Then the authors further have shown that loss of GPR153 expression leads to partial protection from neointima formation in the carotid artery ligation model and reduced severity of neuroinflammation and ischemic damage, which suggests that GPR153 may be a pathological target for parallel modulation of EC inflammation and SMC proliferation upon injuries. Overall, experiments and results presented in this manuscript were done in a relatively meticulous way and support the conclusion in the study, respectively. Furthermore, the present study will bring many attentions in the field of GPCRs and damage-promoted vascular remodeling and inflammatory responses. Nonetheless, there are several major and minor issues to be addressed in this form of the manuscript.

Major

1. Throughout the manuscript, there are a few of descriptions that are confusing to easily follow. The authors are recommended to more explicitly describe them. For example:
 - i. In the summary, lines 42-44, "Mechanistically, we show that GPR153 negatively regulates cellular cAMP levels, resulting in increased CREB phosphorylation, reduced YAP/TAZ levels, and diminished NF- κ B activation in the absence of GPR153." It would be better to be rephrased like "Mechanistically, we show that GPR153 negatively regulates cellular cAMP levels, and thus the absence of GPR153 leads to an increase in CREB phosphorylation, reduced YAP/TAZ levels, and diminished NF- κ B activation".
 - ii. The sentence on lines 264-266. One suggestion would be "Taken together, GPR153 regulates cellular cAMP levels in a Gi/o-independent inhibition of Gs activation", thereby reducing CREB-mediated signaling that inhibits YAP/TAZ and NF- κ B-dependent signals".
 - iii. The sentence on lines 277-278 needs something like "in GPR153 knocked-down cells" at the end, and the sentence on lines 278-279, "upon GPR153 knocking-down" at the end.
 - iv. On the line-313, it needs to add "in GPR153-deficient BECs" after ".....significantly reduced expression of Vcam1,".
2. In the experiment shown in Fig.6P, the authors pretreated cells with PTX for 3h. But the inhibitory effect of PTX often clearly appears after overnight pretreatment. Thus, it is recommended that the authors need to re-test the effect of PTX after overnight pretreatment, particularly for the case with the overexpressed receptor in this study before they draw the current conclusion. Although the authors performed a control experiment for PTX activity with CXCR4 stimulated with SDF1 α , the level of receptor expression and the coupling efficiency to Gi between GPR153 and CXCR4 may be different.
Further, it seems not to be logical that the authors made the conclusion that "GPR153 normally inhibits cAMP production by interfering with G α s-dependent activation of adenylyl cyclases" through the experiment with G α -deficient cells as shown in Fig. 6Q and 6R.
3. On lines 282-284, it is described "Reduced expression of VCAM1 was confirmed on the protein level (Fig. 7M), whereas proliferation, here determined by EdU incorporation, was not significantly altered (Fig. 7N)." The authors need to briefly discuss what will be the reason why the proliferation was not significantly altered even though the expression of proliferation markers shown in Fig. 7L was substantially changed.

4. It would be better if the authors more explicitly describe the conclusion from the adhesion/migration results observed as shown in Fig. 7O-R.
5. The authors described “The line with reduced inflammatory activity of the blood brain barrier,” on the line-315.”. However, it does not seem to be plausible to say this based on the observation shown in Fig. 8F where there is no statistically significant difference in the expression of inflammatory markers except Vcam1 between control and iBEC-G153-KO cells.

Minor

1. In Fig. 3J, it looks like there are some decreases in contractile responses of iSM-G153-KO mice compared to those in control animals unlike what the authors described. If so, it would be better to say "Taken together, inactivation of GPR153 in SMCs does not *substantially* affect basic vascular function, on the line-177.
2. In Fig. 5, the authors may need to briefly describe what was the reason why they examined hCAsMCs more details before AoSMCs to understand how GPR153 regulates SMC function since the previous results were obtained from AoSMCs.
3. On the line-207, “select” must be "selected". Further, is there any rational why the expression of the selected target genes of CREB (Fig. 5J) and NF-κB (Fig. 5O) is changed while the others are not.
4. On the line-210, “NF-κB target genes cells” must be "selected NF-κB target gene”.
5. In Fig. 5P, the authors have shown the reduced TAZ level, how about the extent of YEP as shown in the serum-starvation condition (Fig. 5A and C).
6. On the line-214, it would be better to add "in spite of no statistically significance" after “We found that nuclear MRTF-A localization was in trend increased in GPR153-deficient cells in the starved state,”
7. On the line-219, it would be better to add “"in selected target genes" at the end of the sentence.
8. In the figure legend for Fig. 6K, it would be clearer to change "fold changes" to "PGE2-induced fold changes over basal".
9. On the line-259, “cAMP” would be better with “the cAMP level”.
10. In the figure legend for Fig. 7A. it is described as "data reanalyzed from71". the reference-71 must be the -23 as described in the text.
11. In Fig. 8C, the authors may need to describe what would be the reason while Vcamp1 expression was impaired, Icam1 expression did not altered in GPR153-deficient BECs compared to the control cells?
12. On the line-314, it would be better to add "in spite of no statistical significance" at end of the sentence for better description of the corresponding result shown in Fig. 8F.